



# An Inversion of $NO_x$ and NMVOC Emissions using Satellite Observations during the KORUS-AQ Campaign and Implications for Surface Ozone over East Asia

Amir H. Souri[1][*], Caroline R. Nowlan[1], Gonzalo González Abad[1], Lei Zhu[1,2], Donald R. Blake[3], Alan Fried[4], Andrew J. Weinheimer[5], Jung-Hun Woo[6], Qiang Zhang[7], Christopher E. Chan Miller[1], Xiong Liu[1], and Kelly Chance[1]

[1]Harvard-Smithsonian Center for Astrophysics, Cambridge, MA, USA
[2]School of Environmental Science and Engineering, Southern University of Science and Technology, Shenzhen, China
[3]Department of Chemistry, University of California, Irvine, Irvine, CA, USA
[4]Institute of Arctic & Alpine Research, University of Colorado, Boulder, CO, USA
[5]National Center for Atmospheric Research, Boulder, CO, USA
[6]Department of Advanced Technology Fusion, Konkuk University, Seoul, South Korea
[7]Department of Earth System Science, Tsinghua University, Beijing, China

* corresponding author: ahsouri@cfa.harvard.edu

**Abstract.** The absence of up-to-date emissions has been a major impediment to accurately simulate aspects of atmospheric chemistry, and to precisely quantify the impact of changes of emissions on air pollution. Hence, a non-linear joint analytical inversion (Gauss-Newton method) of both volatile organic compounds (VOC) and nitrogen oxides ($NO_x$) emissions is made by exploiting the Smithsonian Astrophysical Observatory (SAO) Ozone Mapping and Profile Suite Nadir Mapper (OMPS-NM) formaldehyde (HCHO) and the National Aeronautics and Space Administration (NASA) Ozone Monitoring Instrument (OMI) tropospheric nitrogen dioxide ($NO_2$) retrievals during the Korea-United States Air Quality (KORUS-AQ) campaign over East Asia in May-June 2016. Effects of the chemical feedback of $NO_x$ and VOCs on both $NO_2$ and HCHO are implicitly included through iteratively optimizing the inversion. Emissions estimates are greatly improved (averaging kernels>0.8) over medium- to high-emitting areas such as cities and dense vegetation. The amount of total $NO_x$ emissions is mainly dictated by values reported in the MIX-Asia 2010 inventory. After the inversion we conclude a decline in the emissions (before, after, change) for China (87.94±44.09 Gg/day, 68.00±15.94 Gg/day, -23%), North China Plain (NCP) (27.96±13.49 Gg/day, 19.05±2.50 Gg/day, -32%), Pearl River Delta (PRD) (4.23±1.78 Gg/day, 2.70±0.32 Gg/day, -36%), Yangtze River Delta (YRD) (9.84±4.68 Gg/day, 5.77±0.51





Gg/day, -41%), Taiwan (1.26±0.57 Gg/day, 0.97±0.33 Gg/day, -23%), and Malaysia (2.89±2.77 Gg/day, 2.25±1.34 Gg/day, -22%), all of which have effectively implemented various stringent regulations. In contrast, South Korea (2.71±1.34 Gg/day, 2.95±0.58 Gg/day, +9%) and Japan

(3.53±1.71 Gg/day, 3.96±1.04 Gg/day, +12%) experience an increase in $NO_x$ emissions potentially due to risen number of diesel vehicles and new thermal power plants. We revisit the well-documented positive bias of the model in terms of biogenic VOC emissions in the tropics. The inversion, however, suggests a larger growth of VOC (mainly anthropogenic) over NCP (25%) than previously reported (6%) relative to 2010. The spatial variation in both magnitude and sign

of $NO_x$ and VOC emissions results in non-linear responses of ozone production/loss. Due to simultaneous decrease/increase of $NO_x$/VOC over NCP and YRD, we observe an ~53% reduction in the ratio of the chemical loss of $NO_x$ ($LNO_x$) to the chemical loss of $RO_x$ ($RO_2+HO_2$) transitioning toward $NO_x$-sensitive regimes, which in turn, reduces/increases the afternoon chemical loss/production of ozone through $NO_2+OH$ (-0.42 ppbv hr$^{-1}$)/$HO_2$ (and $RO_2$)+NO (+0.31

ppbv hr$^{-1}$). Conversely, a combined decrease in $NO_x$ and VOC emissions in Taiwan, Malaysia, and the southern China suppresses the formation of ozone. Ultimately, model simulations indicate enhancements of maximum daily 8-hour average (MDA8) surface ozone over China (0.62 ppbv), NCP (4.56 ppbv), and YRD (5.25 ppbv) due to the non-linear ozone chemistry, suggesting that emissions standards should be extended to regulate VOCs to be able to curb ozone production

rates. Taiwan, Malaysia, and PRD stand out as the regions undergoing lower MDA8 ozone levels resulting from the $NO_x$ reductions occurring predominantly in $NO_x$-sensitive regimes.



## Introduction

The study of ozone ($O_3$) formation within the troposphere in East Asia is of global importance. This significant pollutant is not confined to the source, as it spreads hemispherically through the air, affecting background concentrations as far away as U.S. A study by Lin et al. [2017] provided modeling evidence of enhancements of springtime surface ozone levels (+0.5 ppbv $yr^{-1}$) in the western U.S. in 1980-2014 solely due to the tripling of Asian anthropogenic emissions over the period. As more studies have informed the impact of ozone pollution on both human health and crop yields, Chinese governmental regulatory agencies have begun to take action on cutting the amount of $NO_x$ ($NO+NO_2$) emissions since 2011-2012 [Gu et al., 2013; Reuter et al., 2014; Krotkov et al., 2016; de Foy et al., 2016; Souri et al., 2017]; however no effective policy on volatile organic compound (VOC) emissions, emitted from various sources such as solvent use, mobile, and chemical industries [Liu et al., 2008a,b], had been put into the effect prior to 2016 [Stavrakou et al., 2017; Souri et al., 2017; Shen et al., 2019; Li et al., 2019], with an exception to Pearl River Delta (PRD) [Zhong et al. 2013]. In addition to China, a number of governments including those of Malaysia and Taiwan have put a great deal of effort into shifting their energy pattern from consuming fossil fuels to renewable sources [Trappey el al., 2012; Chua and Oh, 2011]. On the other hand, using satellite observations, Irie et al. [2016] and Souri et al. [2017] revealed a systematic hiatus in the reduction of $NO_x$ over South Korea and Japan potentially due to increases in the number of diesel vehicles and new thermal power plants built to compensate for the collapse of the Fukushima nuclear power plant in 2011. Therefore, it is interesting to quantify to what extent these policies have impacted ozone pollution.

Unraveling the origin of ozone is complicated by a number of factors encompassing the nonlinearity of ozone formation to its sources, primarily from $NO_x$ and VOCs. Therefore, to be able to quantify the impact of recent emission changes, we have developed a top-down estimate of emission inventories using well-characterized observations. There are a myriad of studies focusing on optimizing the bottom-up anthropogenic and biogenic emissions using satellites observations, which provide high spatial coverage, in conjunction with chemical transport models for VOCs [e.g., Palmer et al., 2003; Shim et al., 2005; Curci et al., 2010; Stavrakou et al., 2009, 2011], and $NO_x$ [Martin et al., 2003; Chai et al., 2009; Miyazaki et al., 2017; Souri et al., 2016a, 2017, 2018]. Most inverse modeling studies do not consider both $NO_2$ and formaldehyde (HCHO) satellite-based observations to perform a joint-inversion. It has been shown that VOC and $NO_x$ emissions





can affect the production/loss of each other [Marais et al., 2012; Wolfe et al. 2016; Valin et al.,
2016; Souri et al., 2020]. Consequently, a joint method that incorporates both species while
minimizing the uncertainties in their emissions is better suited to address this problem. Dealing
with this tangled relationship between VOC-$NO_2$ and $NO_x$-HCHO requires an iteratively non-
linear inversion framework able to incrementally consider the relationships derived from a
chemical transport model. Here we will provide an optimal estimate of $NO_x$ and VOC emissions
during the KORUS-AQ campaign using the Smithsonian Astrophysical Observatory (SAO) Ozone
Mapping and Profile Suite Nadir Mapper (OMPS-NM) HCHO and the National Aeronautics and
Space Administration (NASA) Ozone Monitoring Instrument (OMI) $NO_2$ retrievals whose
accuracy and precisions are characterized against rich observations collected during the campaign.
Having a top-down constraint on both emissions permits a more precise quantification of the
impact of the recent emission changes on different chemical pathways pertaining to ozone
formation and loss.

**Measurements, Modeling and Method**

***Remote sensing measurements***

*OMPS HCHO*

OMPS-NM onboard the Suomi National Polar-orbiting Partnership (Suomi NPP) is a UV-
backscattered radiation spectrometer launched in October 2011 [Flynn et al., 2014]. Its revisit time
is the same as other NASA A-Train satellites, including Aura at approximately 13:30 local time at
the equator in ascending mode. OMPS-NM covers 300-380 nm with a resolution of 1 nm full-
width half maximum (FWHM). The sensor has a 340×740 pixel charge-coupled device (CCD)
array measuring the UV spectra at a spatial resolution of 50×50 $km^2$ at nadir. The HCHO retrieval
has been fully described in González Abad et al. [2015; 2016]. Briefly, OMPS HCHO slant
columns are fit using direct radiance fitting [Chance, 1998] in the spectral range 327.7-356.5 nm.
The spectral fit requires a reference spectrum as function of the cross-track position as it attempts
to determine the number of molecules with respect to a reference (i.e., a differential spectrum
fitting). To account for this, we use earthshine radiances over a relatively clear area in the remote
Pacific Ocean within -30º to +30º latitudes. An upgrade to this reference correction is the use of
daily HCHO profiles over the mean climatological ones from simulations done by the GEOS-
Chem chemical transport model. The scattering weights describing the sensitivity of the light path
through a simulated atmosphere are calculated using VLIDORT. The shape factors used for



calculating air mass factors (AMFs) are derived from a regional chemical transport model (discussed later) that is used for carrying out the inversion in the present study. We remove unqualified pixels based on cloud fraction < 40%, solar zenith angle < 65°, and a main quality flag provided in the data. We oversample the HCHO columns for the period of May-June 2016 using a Cressman spatial interpolator with a 1° radius of influence.

*OMI Tropospheric NO$_2$*

           We use NASA OMI tropospheric NO$_2$ (version 3.1) level 2 data whose retrieval is made in the violet/blue (402-465 nm) due to strong absorption of the molecule in this wavelength range [Levelt et al., 2018]. The sensor has a nadir spatial resolution of 13×24 km$^2$ which can extend to 40×160 km$^2$ at the edge of scanlines. A more comprehensive description of the retrieval and the

uncertainty associated with the data can be found in Krotkov et al. [2017] and Choi et al. [2019]. We remove bad pixels based on cloud fraction < 20%, solar zenith angle < 65°, without the row anomaly, vertical column density (VCD) quality flag = 0, and Terrain Reflectivity < 30%. Similar to the OMPS HCHO, we recalculate AMFs by using shape factors from the chemical transport model used in this study. We oversample the OMI granules using the Cressman interpolator with

a 0.25° radius of influence.

           *Model simulation*

           To be able to simulate the atmospheric composition, and to perform an analytical inverse modeling, we set up a 27-km grid resolution regional chemical transport model using the Community Multiscale Air Quality Modeling System (CMAQ) model [Byun and Schere, 2006]

that consists of 328×323 grids covering China, Japan, South Korea, Taiwan and some portions of Russia, India and South Asia (Figure 1). The time period covered by the simulation is from April to June 2016. We use the month of April for spin-up. The anthropogenic emissions are based on the monthly MIX-Asia 2010 inventory [Li et al., 2015] in the CB05 mechanism. The anthropogenic emissions are mainly grouped into three different sectors, namely mobile, point,

and residential (area) sources. We apply a diurnal scale to the mobile sectors used in the national emission inventory (NEI)-2011 emission platform to represent the first-order approximation of traffic patterns. We include biomass burning emissions from the Fire Inventory from NCAR (FINN) v1.6 inventory [Wiedinmyer et al., 2011], and consider the plume rise parametrization used in the GEOS-Chem model (i.e., 60% of emissions are distributed uniformly in the planetary

boundary layer (PBL)). We use the offline Model of Emissions of Gases and Aerosols from Nature



(MEGAN) v2.1 model [Guenther et al., 2006] following the high resolution inputs described in Souri et al. [2017]. The diurnally lateral chemical conditions are simulated by GEOS-Chem v10 [Bey et al., 2001] using the full chemistry mechanism ($NO_x$-$O_x$-HC-Aer-Br) spun up for a year. With regard to weather modeling, we use the Weather Research and Forecasting model (WRF) v3.9.1 [Skamarock et al., 2008] at the same resolution to that of the CMAQ (~27km), but with a wider grid (342×337), and 28 vertical pressure sigma levels. The lateral boundary conditions and the grid nudging inputs are from the global Final (FNL) 0.25º resolution model. The major configurations for the WRF-CMAQ model are summarized in Table 1 and Table 2.

*Inverse modeling*

We attempt to improve our high-dimensional imperfect numerical representation of atmospheric compounds using the well-characterized $NO_2$ and HCHO columns from satellites. We use an analytical inversion using the WRF-CMAQ model to constrain the relevant bottom-up emission estimation [Souri et al., 2016; Souri et al., 2017; Souri et al., 2018]. The inversion seeks to solve the following cost function under the assumptions that i) both observation and emission error covariances follow Gaussian probability density functions with a zero bias, ii) the observation and emission error covariances are independent and iii) the relationship between observations and emissions is not grossly non-linear:

$$J(\mathbf{x}) = \frac{1}{2}(\mathbf{y} - F(\mathbf{x}))^T \mathbf{S}_o^{-1}(\mathbf{y} - F(\mathbf{x})) + \frac{1}{2}(\mathbf{x} - \mathbf{x}_a)^T \mathbf{S}_e^{-1}(\mathbf{x} - \mathbf{x}_a) \tag{1}$$

where $\mathbf{x}$ is the inversion estimate (a posteriori) given two sources of data: a priori ($\mathbf{x}_a$) and observation ($\mathbf{y}$). $\mathbf{S}_o$ and $\mathbf{S}_e$ are the error covariance matrices of observation (instrument) and emission. $F$ is the forward model (here WRF-CMAQ) to project the emissions onto columns. The first term of Eq.1 attempts to reduce the distance between observations and the simulated columns. The second term incorporates some prior understanding and expectation of the true state of the emissions, that is, it does not allow the a posteriori to deviate largely from the a priori, even though the observations could be far from our estimation. The weight of each term is dictated by its covariance matrix. If $\mathbf{S}_e$ is large compared to $\mathbf{S}_o$, the a posteriori will be independent of the prior knowledge and, conversely, if $\mathbf{S}_o$ dominates, the final solution will consist mostly of the a priori.

Following the Gauss-Newton method described in Rodger [2000], we derive iteratively (i.e., $i$ is the index of iteration) the posterior emissions by:

$$\mathbf{x}_{i+1} = \mathbf{x}_a + \mathbf{G}[\mathbf{y} - F(\mathbf{x}_i) - K_i(\mathbf{x}_i - \mathbf{x}_a)] \tag{2}$$



where **G** is the Kalman gain,

$$\mathbf{G} = \mathbf{S}_e \, K_i^T \left( K_i \mathbf{S}_e \, K_i^T + \mathbf{S}_o \right)^{-1} \tag{3}$$

and $K_i \, (= K(\mathbf{x}_i))$ is the Jacobian matrix calculated explicitly from the model (discussed later). The covariance matrix of the a posteriori is calculated by:

$$\hat{\mathbf{S}}_e = (\mathbf{I} - \mathbf{G}\hat{K}^T)\mathbf{S}_e \tag{4}$$

where $\hat{K}$ is the Jacobian from the i*th* iteration. Here we iterate Eq.2 three times. The averaging kernels (**A**) are given by:

$$\mathbf{A} = \mathbf{I} - \hat{\mathbf{S}}_e \mathbf{S}_e^{-1} \tag{5}$$

The inversion system is complicated by the commonly overlooked fact that observations are biased. For instance, Souri et al. [2018] found that airborne remote sensing observations were high relative to surface Pandora measurements. The overestimation of the VCDs was problematic, since it could have been propagated in the inversion, inducing a bias in the top-down estimation. The authors partly mitigated it by constraining the MODIS albedo which was assumed to be responsible for the bias. Attempts to reduce the bias resulting from coarse profiles from a global model in calculating gas shape profiles were made by recalculating the shape factors using those from higher spatial resolution regional models in other studies [e.g., Souri et al., 2017; Laughner et al., 2018]. For this study, we use abundant observations from the KORUS-AQ campaign and follow the intercomparison platform proposed by Zhu et al. [2016; 2020] using aircraft observations collected during the campaign to be able to mitigate the biases in HCHO columns. Based on the corrected global model as a benchmark, we scale up all OMPS HCHO columns by 20%. To mitigate the potential biases in OMI $NO_2$, we followed exclusively the values reported over the KORUS-AQ period in Choi et al. [2019]. We increase the $NO_2$ concentration uniformly by 33.9% (see table A3 in the paper).

We calculate the covariance matrix of observations using the column uncertainty variable provided in the satellite datasets and consider them as random errors. Therefore, these values are significantly lowered down by oversampling the data over the course of two months. In addition to that, we take into account a fixed error for all pixels due to variability that exists in the applied bias correction. This error is based on the RMSE obtained from the mentioned studies used for removing biases.





To increase the degree of freedom for the optimization, we combine all sector emissions including anthropogenic, biomass burning and biogenic emissions for $NO_x$ and VOCs. Therefore, we use the following formula to estimate the covariance of the a priori:

$$\sigma_{Total}^2 = f_{Anthro}^2 \times \sigma_{Anthro}^2 + f_{BB}^2 \times \sigma_{BB}^2 + f_{Bio}^2 \times \sigma_{Bio}^2 \qquad (6)$$

where $f$ denotes the fraction of the emission sector with respect to the total emissions, and $\sigma$ is the standard deviation of each sector category which is calculated from the average of each sector to

a relative error listed in Table 3.

For the same purpose (enhancing the amount of information gained from satellite observation) and to increase computational speed, we reduce the dimension of the state vectors (emissions) by aggregating them. However, grouping emissions into certain zones could also introduce another type of uncertainty, known as the aggregation error. We choose optimally

aggregated zones by running the inversion multiple times, each with a certain selection of state vectors [Turner and Jacob, 2015]. As in our previous study in Souri et al. [2018], we use the Gaussian Model Mixture (GMM) method to cluster emissions into certain zones that share roughly similar features and investigate which combinations will lead to a minimum of the sum of aggregation and smoothing errors.

In order to create the $K$ matrix, one must estimate the impact of changes in emissions for each of the aggregated zones to the concentrations of a target compound which is calculated using CMAQ-Direct Decoupled Method (DDM) [Dunker et al., 1989; Cohan et al., 2005]. For instance, the first row and column of $K$ denoting the response of the first grid cell to a zonal emission can be obtained by:

$$K_{(1,1)} = \frac{S_{(1,1)}^{NO2}}{ENO_x^{Total,Zone}} \qquad (7)$$

where $S_{(1,1)}^{NO2}$ is the sensitivity result in units of molecule cm$^{-2}$ for the first grid indicating how concentration of $NO_2$ column will change at the first row and column of the domain by changing one unit of emission of total $NO_x$ emissions. We do not consider the interconnection between the zonal emissions and concentrations due to computational burdens. The same concept will be applied to HCHO and VOC emissions. The advantage of using CMAQ-DDM to estimate the

sensitivity lies in the fact that it calculates the local gradient which better represents the non-linear relationship existing between the emissions and the columns [Souri et al., 2017; Souri et al., 2018], which in turn, reduces the number of iterations.





**Validation of the model in terms of meteorology**

It is essential to first evaluate some key meteorological variables, because large errors in the weather can complicate the inversion [e.g., Liu et al. 2017]. In order to validate the performance of the WRF model in terms of a number of meteorological variables including surface temperature, relative humidity, and winds, we use more than 1100 surface measurements from integrated surface database (ISD) stations (https://www.ncdc.noaa.gov/isd) over the domain in May-June 2016. Table 4 lists the comparison of the model and the observations for the mentioned variables. Our model demonstrates a very low bias (0.6°C) with regard to surface temperature. We find a reasonable correspondence in terms of relative humidity indicating a fair water vapor budget in the model. The largest discrepancy between the model and observations in terms of temperature and humidity occurs in those grid cells that are in the proximity of the boundary conditions (not shown). Concerning the wind components, the deviation of the model from the observations is smaller than results obtained in a relatively flat area like Houston in Souri et al. [2016].

**Comparison of the model and the satellite observations**

Prior to updating the emissions, we find it necessary to shed light on the spatial distribution of tropospheric $NO_2$ and HCHO total columns from both observations and model, and their potential differences relative to their key precursors emissions.

*$NO_2$*

The first row in Figure 2 illustrates tropospheric $NO_2$ columns from the regional model, OMI (using adjusted AMF and bias corrected), and the logarithmic ratio of both quantities in May-June 2016 at ~1330 LST over Asia. The second row depicts daily-mean values of dominant sources of $NO_x$, namely as, biogenic, anthropogenic, and biomass burning emissions (that are subject to change after the inversion). A high degree of correlation between the anthropogenic $NO_x$ emissions and $NO_2$ columns implies the predominant production of $NO_2$ from the anthropogenic sources [Logan, 1983]. We find a reasonable two-dimensional Pearson correlation ($r$=0.73) between the modeled and the observed columns. Generally, the WRF-CMAQ largely underestimated (56%) tropospheric $NO_2$ columns with respect to those of OMI over the entire domain. Segregating intuitively the domain into high emission areas ($NO_x$ > 10 ton/day) and low ones ($NO_x$ < 10 ton/day) allows for a better understanding of the discrepancy between the model and the observations. In the high $NO_x$ areas, the model tends to overestimate tropospheric $NO_2$ columns by 73%, whereas for the low $NO_x$ regions, the model shows a substantial underestimation by 68%.





Such a conflicting bias is confirmed by the contour map of the logarithm ratio of the model to OMI

in Figure 2. The large overestimation of the model in terms of $NO_2$ over the polluted areas is explained by stringent regulations enacted in various countries in Asia; for instance, Chinese regulatory agencies have taken aggressive actions recently to cut anthropogenic $NO_x$ emissions by implementing selective catalytic reduction in power plants, closing a number of coal power plants, and policies on transportation [Zhang et al., 2012, Liu et al., 2016]. The highest positive bias in

the model is observed over Shanxi Province in China, home to coal production, underscoring the effectiveness of the emission standards at controlling air pollution. Likewise, we observe a positive bias in the model over major cities in Japan and South Korea; but the magnitude of the reduction over these cities is substantially smaller than what we observe in China. In particular, Irie et al. [2016] and Souri et al. [2017] found a hiatus in the $NO_x$ reduction over Japan and South Korea

during the 2010-2014 period mainly due to rapid increases in the number of diesel cars in South Korea, and thermal power plants built as a substitution for the Fukushima nuclear plant in Japan.

The underestimation of the model in the low $NO_x$ regions is related to a number of factors such as i) the widely-reported underestimation of soil (biogenic) $NO_x$ emissions due to the lack of precise knowledge of fertilizers use, soil biota, or canopy interactions [Jaeglé, et al., 2005; Hudman

et al., 2010; Souri et al., 2016], ii) the underestimation of the upper-troposphere $NO_2$ due to non-surface emissions (aviation/lightning) or errors in the vertical mixing or moist convection [e.g., Souri et al., 2018], and iii) a possible overprediction of the lifetime of organic nitrates diminishing background $NO_2$ levels [Canty et al., 2015]. Addressing the second issue requires a very high resolution model with explicit resolving microphysics and large eddy simulations, and the last

problem requires more experimental studies to improve organic nitrates chemistry [Romer Present et al., 2020]. In this study, we attempt to mitigate the discrepancy between the model and the satellite observations solely by adjusting the relevant emissions. Accordingly, future approaches improving models the physical/chemical processes can offset top-down emissions estimates inevitably.

***HCHO***

A comparison between HCHO columns from the model and OMPS along with the major sources of VOCs in May-June 2016 is depicted in Figure 3. A reasonable correlation (*r*=0.78) between the model and OMPS suggests a good confidence in the location of emissions. However, the magnitude of HCHO columns between the two datasets strongly disagrees, especially over the



tropics where biogenic emissions are large. A myriad of studies have reported a largely positive
bias (by a factor of 2-3) associated with isoprene emissions estimated by MEGAN using satellite
measurements [e.g., Millet et al., 2008; Stavrakou et al., 2009; Marais et al., 2012; Bauwens et al.,
2016]. To compound, Stavrakou et al. [2011] found a large overestimation in methanol emissions
from the same model that can further preclude the accurate estimation of the yield of HCHO. This
is especially the case for the tropics. As a response to the overestimation of the biogenic VOCs by
MEGAN, we observe a largely positive bias in the simulated HCHO columns ranging from 50%
over the south of China to ~400% over Malaysia and Indonesia. As we move away from the hotspot
of the biogenic emissions in lower latitudes, the positive bias of the model declines, ultimately
turning into a negative bias at higher latitudes. OMPS HCHO columns suggest that the yield of
HCHO over North China Plain (NCP) and Yangtze River Delta (YRD) is comparable to those over
the tropics suggesting that the anthropogenic emissions over NCP are the dominant source of
HCHO [Souri et al., 2017; Jin and Holloway, 2015]. We do not see a significant deviation in the
model from the observations over this region indicating that no noticeable efforts on controlling
VOC emissions in NCP and YRD have been made which is very likely due to the fact that the
recent regulations over China have overlooked cutting emissions from several industrial sectors
[Liu et al., 2016] prior to 2016 [Li et al. 2019]. This finding lines up with results reported by Souri
et al. [2017] and Shen et al. [2019]. We observe both underestimated and overestimated values in
the simulated HCHO columns over areas in South Korea and Japan. The underestimation of HCHO
in the model over regions with low VOCs (such as Mongolia and Pacific Ocean) can be either due
to missing sources or the incapability of CMAQ to account for moist convective transport. As
shown here, it is necessary to adjust the emissions to better match the simulated columns with the
satellites observations given their errors, and by doing so, there is a chance for a better simulation
of the formation of tropospheric ozone.

**Updated Emissions**

In this section we report the results from the inverse modeling and the associated
uncertainty associated with the top-down estimation; moreover, we wish to assess how much
information is gained from utilizing satellite observations via the calculation of averaging kernels.
Finally, observations are used to verify, to some extent, the accuracy of our top-down emission
estimations.

*$NO_x$*


The first row in Figure 4 shows the a priori, the a posteriori, and their ratios in terms of the total $NO_x$ emissions in May-June 2016. We observe that the ratios are highly correlated with those of CMAQ/OMI shown in Figure 2, suggesting that the inversion attempts to reduce the distance between the model and the observations. Major reductions occur over China. We attribute them to

strict emissions policies [Liu et al., 2016; Reuter et al., 2015; de Foy et al., 2016; Krotkov et al., 2016; Souri et al., 2017]. The enhancements in $NO_x$ emissions are commonly found in rural areas, especially over grasslands located in the western/central China and Mongolia. The changes in $NO_x$ emissions over South Korea and Japan are positive [Irie et al., 2016; Souri et al., 2017]. This is especially the case for Japan for which we observe a larger enhancement in total $NO_x$ emissions

(12%) essentially due to new thermal power-plants. The second row in Figure 4 depicts the relative errors in the a priori, the a posteriori, and AKs. Relative errors in the a priori are mostly confined to values close to 50% in polluted areas. They increase further, up to 100%, in areas experiencing relatively large contributions from biomass burning or biogenic (soil) emissions. Encouragingly, OMI tropospheric $NO_2$ columns in conjunction with the solid mathematical inversion method

[Rodger, 2000] greatly reduce the uncertainties associated with the emissions in polluted areas; we observe AKs close to 1 over major cities or industrial areas. We see the lowest values in AKs over rural areas due to weaker signal/noise ratios from the sensor. Therefore, it is desirable but very difficult to improve the model using the sensor in terms of $NO_x$ chemistry/emissions in remote areas, evident in the low values of AKs. Table 5 lists the magnitude of the total $NO_x$ emissions in

several regions (refer to Figure 1) before and after carrying out the inversion. If we assume that the dominant source of $NO_x$ emissions is anthropogenic, the most successful countries at cutting emissions (before, after) are China (87.94±44.09 Gg/day, 68.00±15.94 Gg/day), Taiwan (1.26±0.57 Gg/day, 0.97±0.33 Gg/day), and Malaysia (2.89±2.77 Gg/day, 2.25±1.34 Gg/day). All three countries have successfully implemented plans to reduce anthropogenic emissions since

2010-2011 [Zhang et al., 2012; Trappey el al., 2012; Chua and Oh, 2011]. The uncertainty associated with the top-down estimate improves considerably. The largest reduction in the uncertainty of the emissions is observed over China, a response to a strong signal from OMI.

An interesting observation lies in the discrepancy between the ratio of OMI/CMAQ (Figure 2) to that of the a posteriori to the a priori over the North China Plain, suggesting that using a bulk

ratio [Martin et al., 2003] cannot fully account for possible chemical feedback. The ratio of OMI/CMAQ is consistently lower than changes in the emission. Two reasons contribute to this



effect: i) as $NO_x$ emissions decrease in $NO_x$-saturated areas (i.e., the dominant sink of radicals is through $NO_2+OH$), OH levels essentially increase resulting in a shorter lifetime in $NO_2$; therefore to reduce $NO_2$ concentrations, a substantial reduction in $NO_x$ (suggested by OMI/CMAQ) is

unnecessary coinciding with results from the inverse modeling, ii) the CMAQ-DDM (Figure S1) suggests that $NO_2$ columns decrease due to increasing VOC emissions over the region; accordingly, the cross-relationship between $NO_2$ concentrations and VOC emissions which has been implicitly taken into consideration by iteratively optimizing the cost function partly adds to the discrepancy. It is because of the chemical feedback that recent studies have attempted to

enhance the capability of inverse modeling by iteratively adjusting relevant emissions [e.g., Cooper et al., 2017; Li et al., 2019].

To assess the resulting changes in the tropospheric $NO_2$ columns after the inversion, and to validate our results, we compare the simulated values using the a priori and the a posteriori with OMI in Figure 5. We observe 64% reduction in the tropospheric $NO_2$ columns on average over

NCP despite only 32% reduction in the total $NO_x$ emissions over the region, a result of the chemical feedback. The two-dimensional Pearson correlation between the simulation using the a posteriori and OMI increases from 73% (using the a priori) to 83%. Both datasets now are in a better agreement as far as the magnitude goes. However, we do not see a significant change in the background values in the new simulation compared to those of OMI. This is primarily because of

the consideration of higher covariances over low-emitting areas that weighs up the inversion towards the prior values.

To further validate the results, we compare the $NO_2$ data from the NCAR's four-channel chemiluminescence instrument onboard the DC-8 aircraft during the campaign (not shown). These data are not interfered by $NO_z$ family. The aircraft collected the data in the Korean Peninsula

around 23 days in May-June 2016 covering various altitudes and hours (https://www-air.larc.nasa.gov/cgi-bin/ArcView/korusaq, access date: December 2019). We observe an underestimation of $NO_2$ at the near surface levels (<900 hPa) by 19% (DC8 = 4.50 ppbv, CMAQ = 3.67 ppbv). The updated emissions increase the near surface levels over the Korean Peninsula, which in turn, reduce the bias to 11% (CMAQ = 4.02 ppbv).

***VOC***

Figure 6 illustrates the total VOC emissions before and after the inversion along with their errors. Immediately apparent is the large reduction of VOC emissions in the tropics and subtropics



due to the overestimation of isoprene from MEGAN v2.1. In contrast, enhancements of the emissions are evident at higher latitudes. We observe that the dominantly anthropogenic VOC

emissions over NCP increase (~25%) after the adjustment highlighting the minimal efforts made to reduce this particular source of emissions [Souri et al., 2017; Shen et al., 2019]. For instance, Stavrakou et al. [2017] reported ~6% increases in anthropogenic VOC emissions over China from 2010 to 2014. Despite the presence of vegetation over Japan and South Korea, we do not see largely overestimated values in the emissions. Hence, the overestimation of isoprene emissions is

more pronounced in the tropics possibly because of an overestimation in the emission factors used for specific plants. Nevertheless, a non-trivial oversight in models could be an insufficient representation of both $HO_x$ chemistry and dry deposition in forest canopies [Millet et al., 2008]; as a result, the net amount of HCHO in the atmosphere over forest areas is higher than what should be if removal through either a chemical loss or a faster dry deposition is considered.

Owning to the fact that we assume anthropogenic VOC emissions to be less uncertain relative to other sectors, the errors in the a priori are smaller in populated areas. We observe that OMPS HCHO columns are able to significantly reduce the uncertainty associated with the total VOC emissions over areas showing a strong HCHO signal ($>10^{16}$ molec.cm$^{-2}$). Over clean areas, it is the other way around; we see less confidence in our top-down estimate (AK<0.4) in areas such

as Tibet and Mongolia.

We then compare the simulated HCHO column using two different emission inventories with those of OMPS in Figure 7. We observe a substantial improvement both in the spatial structure and the magnitude of simulated HCHO columns using the a posteriori with respect to OMPS. The two-dimensional Pearson correlation increases from 0.78 to 0.91 after applying the

adjustments to the emissions. In response to the increases in the total VOC emissions over the North China Plain, we observe ~11% enhancements in the simulated HCHO total columns. The updated emissions lead to a reduction in HCHO total columns as large as 70% in the tropics.

Validation of the model in terms of VOCs is not a straightforward task because the chemical mechanism used for our model has lumped several VOC species such as terminal/internal

olefin or paraffin, only a handful of which were measured during the campaign. Besides, the MIX-Asia inventory estimates the anthropogenic emissions for a selected number of VOCs in the CB05 mechanism. Here, we focus only on six compounds including isoprene, HCHO, ethene, ethane, acetaldehyde, and methanol whose emissions are adjusted (with the same factor) based on satellite



measurements. The comparison of the simulated values with the DC-8 measurements showed a
noticeable mitigation in the discrepancy between two datasets at lower boundaries (<900 hPa) in
terms of isoprene (Figure S2), ethane (Figure S3), ethene (Figure S4), and acetaldehyde (Figure
S5). Surprisingly, we observe a large underestimation of methanol over the Korean Peninsula by
a factor of ten (Figure S6). Same tendency was observed in other regions in Wells et al. [2014]
(see Figure 8 in the paper). Our inversion obviously fails at mitigating the bias as there is not much
direct information from the satellite observations on this compound. Wells et al. [2014] and Hu et
al. [2011] demonstrated that methanol can be a secondary source of HCHO up to 10-20% in
midlatitudes in warm seasons. We tend to underestimate HCHO concentrations (by 15%) in the
lower atmosphere (<900 hPa) after using the a posteriori over the Korean Peninsula (Figure S7).

**Implications for surface ozone**

The results we have generated can be further exploited to elucidate changes in the ozone
production rates $P(O_3)$ owing to have constrained both $NO_x$ and VOC emissions. We calculate
$P(O_3)$ by subtracting the ozone loss driven by $HO_x$ ($HO+HO_2$), reaction with several VOCs (i.e.,
alkenes and isoprene), the formation of $HNO_3$, and $O_3$ photolysis followed by the reaction of $O(^1D)$
with water vapor, from the ozone formation via removal of NO through $HO_2$ or $RO_2$:

$$\begin{aligned} P(O_3) = k_{HO_2+NO}[HO_2][NO] + \sum k_{RO_{2i}+NO}[RO_{2i}][NO] \\ - k_{OH+NO_2+M}[OH][NO_2][M] - k_{HO_2+O_3}[HO_2][O_3] \\ - k_{OH+O_3}[OH][O_3] - k_{O(^1D)+H_2O}[O(^1D)][H_2O] - L(O_3 + VOCs) \end{aligned} \tag{8}$$

Since $P(O_3)$ is a non-linear function of $NO_x$ and VOC emissions, it is advantageous to look at the
ratio of chemical loss of $NO_x$ to that of $RO_x$ ($RO_2+HO_2$), a robust indicator to pinpointing
underlying drivers for $RO_x$ cycle. $LRO_x$ is defined through the sum of primarily radical-radical
reactions:

$$LRO_x = k_{HO_2+HO_2}[HO_2]^2 + \sum k_{RO_{2i}+HO_2}[RO_{2i}][HO_2] + \sum k_{RO_{2i}+RO_{2i}}[RO_{2i}]^2 \tag{9}$$

$LNO_x$ mainly occurs via the $NO_2+OH$ reaction:

$$LNO_x = k_{OH+NO_2+M}[OH][NO_2][M] \tag{10}$$

Typically, a value of $LNO_x/LRO_x \sim 2.7$ defines the transition line between VOC-sensitive and
$NO_x$-sensitive regimes [Schroeder et al., 2017; Souri et al., 2020].

Figure 8 depicts a contour map of $LNO_x/RO_x$ ratios before and after the inversion. As
expected, the larger ratios are confined within major cities or industrial areas due to abundant $NO_x$





emissions. The hotspot of VOC-sensitive regimes is located in NCP and YRD. Also of interest in
Figure 8 is that advection renders a major fraction of the Yellow Sea (the sea connecting China to
Korea) VOC-sensitive. Using the a posteriori leads to precipitous changes in the chemical
condition regimes. As a result of a large reduction in the isoprene emissions in both the tropics and
subtropics, we observe a shift toward VOC-limited, though the values of $LNO_x/RO_x$ are yet too
far from the transition line (i.e., <<2.7). The substantial reduction in $NO_x$ emissions and an increase
in VOC emissions over NCP and YRD go hand-in-hand transitioning towards $NO_x$-sensitive
regime. The ratios over South Korea and Japan are found to be variable and somehow in synch
with the changes in $NO_x$ emissions.

The resultant changes in the $LNO_x/LRO_x$ ratios shed some light on ozone sensitivity with
respect to its major precursors, but $P(O_3)$ is also dependent on the absolute values of emissions,
the degree of reactivity of VOCs, and the abundance of radicals. Here we use the integrated
reaction rates (IRR) to determine the most influential reactions pertaining to ozone loss and
production at the surface. We focus on 1200 to 1800 China standard time (CST) hours. Figure 9
shows the differences in the major pathways for the loss and the formation of ozone at the surface
within the time window. The differences are computed based on the subtraction of the simulation
with the a posteriori from that with the a priori. In Figure 9 we see a strong degree of correlation
between the changes in magnitude of $P(O_3)$ through $HO_2+NO$ reaction with those of $NO_x$
emissions (Figure 4), whereas the changes in magnitude of $P(O_3)$ via $RO_2+NO$ reaction primarily
are par with those of VOC emissions (Figure 6). We observe $P(O_3)$ increases through $HO_2+NO$
and $RO_2+NO$ reactions in Japan, South Korea, Myanmar, and Philippines because of increases in
$NO_x$ emissions in $NO_x$-sensitive regions. The simultaneous decrease in $NO_x$ and VOC in PRD and
Taiwan causes the production of ozone via the same pathways to reduce.

Normally, in VOC-rich environments, reduction in VOC emissions boosts OH
concentrations (Figure S8). Consequently, we observe an enhancement in $NO_2+OH$ reaction in the
tropics and subtopics. A substantial reduction in the chemical loss of ozone through $NO_2+OH$ over
NCP and YRD arises from a considerable decrease of $NO_x$ emissions and an increase in OH (due
to chemical feedback of $NO_x$). Because of increases in $HO_x$ concentrations over NCP (Figure S8-
S9), we also observe an enhancement in ozone loss through reacting with $HO_x$. Changes in the
ozone photolysis ($O^1D+H_2O$) are majorly dictated by photolysis and water vapor mixing ratios,
both of which are roughly constant in both simulations; accordingly the difference of the reaction



rate is mainly reflecting those in ozone (shown later). Interestingly, we observe a large reduction in the loss of ozone through reaction with VOCs at lower latitudes. This is essentially because of the reduction in ISOP+$O_3$, a VOC that prevails in those latitudes. Despite a much slower reaction rate for ISOP+$O_3$ compared to ISOP+OH and ISOP+$hv$ [Karl et al. 2004], this specific chemical pathway can be important as a way to oxidize isoprene and forming $HO_x$ in forests [Paulson and

Orlando, 1996].

     Figure 10 sums the differences of all mentioned chemical pathways involved in formation/loss of surface ozone at 1200-1600 CST. Because of a complex non-linear relationship between $P(O_3)$ and its precursors, we observe a variability in both the sign and amplitude of $P(O_3)$. On average, changes in $O_3$ production dominate over changes in $O_3$ sinks except in Malaysia

which underwent a significant reduction in isoprene emissions, thus slowing down the ISOP+$O_3$ reaction. Following the patterns of $NO_x$-limited and VOC-limited in Figure 8, it is possible to conclude that the $P(O_3)$ differences are mainly driven by those of $NO_x$ depending at which chemical condition the changes in emissions have occurred.

     Much of the above analysis is based on ozone production rates, however, various

parameters encompassing dry deposition, vertical diffusion, and advection can also affect ozone concentrations. Therefore we further compute the difference between the simulated maximum daily 8-h average (MDA8) surface ozone levels before and after the inversion depicted in Figure 11. We see a striking correlation between $P(O_3)$ (right panel in Figure 10) and MDA8 surface ozone indicating that the selected chemical pathways in this study can explain ozone changes.

Nonetheless, the transport obviously plays a vital role in the spatial variability associated with the differences of surface ozone [e.g., Souri et al., 2016b]. Figure 11 suggests a significant enhancement of ozone over NCP (~4.56 ppbv, +5.6%) and YRD (5.2 ppbv, +6.8%) due to simultaneous decreases/increases in $NO_x$/VOCs which is in agreement with Li et al. [2019]. On the other hand, reductions in $NO_x$ mitigate ozone pollution in PRD (-5.4%), Malaysia (-5.6%) and

Taiwan (-11.6%). Table 6 lists the simulated MDA8 surface ozone levels for several regions before and after updating the emissions. Increases in MDA8 ozone over NCP and YRD overshadow decreases in southern China resulting in 1.1% enhancement for China. This provides strong evidence that regulations on cutting VOC emissions should not be ignored. The largest reduction/increase of MDA8 ozone is found over Taiwan/YRD.




## Summary

In this paper we have focused on providing a top-down constraint on both volatile organic compound (VOC) and nitrogen oxides ($NO_x$) emissions using a combination of error-characterized Smithsonian Astrophysical Observatory (SAO) Ozone Mapping and Profile Suite Nadir Mapper (OMPS-NM) formaldehyde (HCHO) and National Aeronautics and Space Administration (NASA) Ozone Monitoring Instrument (OMI) nitrogen dioxide ($NO_2$) retrievals during the Korean and United States (KORUS) campaign over East Asia in May-June 2016. Here, we include biogenic, biomass burning and anthropogenic emissions from MEGAN, FINN, and MIX-Asia 2010 inventory, respectively. A key point is that by considering together the satellite observations, we have been able to not only implicitly take the chemical feedback existing between HCHO-$NO_x$ and $NO_2$-VOC into account through iteratively optimizing an analytical non-linear inversion, but also to quantify the impact of recent changes in emissions (since 2010) on surface ozone pollution.

Concerning total $NO_x$ emissions, the inversion estimate suggests a substantial reduction over China (-23%), North China Plain (NCP) (-32%), Pearl River Delta (PRD) (-36%), Yangtze River Delta (YRD) (-41%), Taiwan (-23%), and Malaysia (-22%) with respect to the values reported in the prior emissions mostly dictated by the MIX-Asia 2010 inventory. In essence these values reflect recent actions to lower emissions in those countries [Zhang et al., 2012; Trappey el al., 2012; Chua and Oh, 2011]. The analytical inversion also paves the way for estimating the averaging kernels (AKs), thereby informing the amount of information acquired from satellites on the emissions estimation. We observe AKs>0.8 over major polluted areas indicating that OMI is able to improve the emission estimates over medium to high-emitting regions. Conversely, AKs are found to be small over pristine areas suggesting that little information can be gained from the satellite over rural areas given retrieval errors. In line with the studies of Irie et al. [2016] and Souri et al. [2017], we observe a growth in the total $NO_x$ emissions in Japan (12%) and South Korea (+9%) which are partially explained by construction of new thermal power plants in Japan, and an upward trend in the number of diesel vehicles in South Korea.

MEGAN v2.1 estimates too much isoprene emissions in the tropics and subtropics, a picture that emerges from the latitudinal dependence of the posterior VOC emissions to the prior ones. It is readily apparent from the top-down constrained VOC emissions that the prevailing anthropogenic VOC emissions in NCP is underestimated by 25%, a direction that is in agreement with studies by Souri et al. [2017] and Shen et al. [2019]. We find out that OMPS HCHO columns





can greatly reduce the uncertainty associated with the total VOC emissions (AKs>0.8) over regions having a moderate-strong signal ($>10^{16}$ molec.cm$^{-2}$).

A large spatial variability associated with both $NO_x$ and VOC results in great oscillation in chemical conditions regimes (i.e., $NO_x$-sensitive or VOC-sensitive). Due to considerable reduction/increase in NOx/VOC emissions in NCP and YRD, we observe a large increase (53%) in the ratio of the chemical loss of $NO_x$ ($LNO_x$) to the chemical loss of $RO_x$ ($RO_2+HO_2$) shifting the regions towards $NO_x$-sensitive. As a result, a substantial reduction in afternoon $NO_2+OH$ reaction rate (a major loss of $O_3$), and an increase in afternoon $NO+HO_2$ and $RO_2+NO$ (a major

production pathway for $O_3$) are observed, leading to enhancements of the simulated maximum daily 8-hr average (MDA8) surface ozone concentrations by ~5 ppbv. Therefore, additional regulations on VOC emissions should be implemented to battle ozone pollution in those areas. On the other hand, being predominantly in $NO_x$-sensitive regimes favors regions including Taiwan, Malaysia and PRD to benefit from reductions in $NO_x$, resulting in noticeable decreases in

simulated MDA8 surface ozone levels.

It has taken many years to develop satellite-based gas retrievals, and weather and chemical transport models accurate enough to enable observationally-based estimates of emissions with reasonable confidence and quantified uncertainty, and produce credible top-down emission inventories over certain areas. However it is essential to improve certain aspects to be able to

narrow the range of uncertainty associated with the estimation: i) getting the bias of the satellite gas retrievals about right for some areas (which requires a rigorous construction of representivity factor when it comes to comparing two datasets) would be insufficient because the bias can vary substantially over time and space depending on underlying surface properties and the atmospheric state, ii) there is a need for a proper quantification of the errors in the prior emissions based on the

very raw information used in the emission inventory to discourage some arbitrarinesses, iii) the model parameter errors including those from PBL, radiation, and winds should be propagated to the final output [e.g., Rodger 2000], iv) due to intertwisted chemical feedback between various chemical compounds, inverse modeling needs to properly incorporate all available information (beyond HCHO and $NO_2$) considering the cross-relationship either explicitly or implicitly.

**Acknowledgment**

We are thankful for the funding from NASA Aura Science Team (#NNX17AH47G), and NOAA AC4 program (#NA18OAR4310108). We acknowledge the publicly available WRF, CMAQ,





GEOS-Chem models, and KORUS-AQ data that make this study possible. The simulations were run on the Smithsonian Institution High Performance Cluster (SI/HPC).

**Authors' contributions**

A.H.S designed the research, analyzed the data, conducted the inverse modeling, CMAQ, GEOS-Chem, WRF, and MEGAN, made all figures and wrote the manuscript. C.R.N, G.G, C.E.C.M, X.L. and K.C retrieved OMPS HCHO columns and conceived the study. L.Z. validated OMPS HCHO. D.R.B, A.F, and A.J.W measured different compounds during the campaign. J.W and Q.Z

provided MIX-Asia inventory. All authors contributed to discussions and edited the manuscript.




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





**Table 1.** CMAQ major configurations

| CMAQ version | V5.1 |
|---|---|
| Chemical Mechanism | CB05 with chlorine chemistry |
| Lightning $NO_x$ emission | Included using inline code |
| Photolysis | Inline including aerosol impacts |
| Horizontal advection | YAMO (hyamo) |
| Vertical advection | WRF omega formula (vwrf) |
| Horizontal mixing/diffusion | Multiscale (multiscale) |
| Vertical mixing/diffusion | Asymmetric Convective Model version 2 (acm2) |
| Aerosol | AERO 6 for sea salt and thermodynamics (aero6) |
| IC/BC source | GEOS-Chem v10 |

**Table 2.** WRF physics options

| WRF Version | V3.9.1 |
|---|---|
| Microphysics | WSM-6 |
| Long-wave Radiation | RRTMG |
| Short-wave Radiation | RRTMG |
| Surface Layer Option | Monin-Obukhov |
| Land-Surface Option | Noah LSM |
| Boundary Layer | ACM2 |
| Cumulus Cloud Option | Kain-Fritsch |
| IC/BC | FNL 0.25º |


**Table 3.** The uncertainty assumptions used for estimating the covariance matrix of the a priori.

|  | Anthropogenic | Biogenic | Biomass Burning |
|---|---|---|---|
| $NO_x$ | 50% | 200% | 100% |
| VOC | 150% | 200% | 300% |

**Table 4.** Statistics of surface temperature, relative humidity, and wind. Corr – Correlation;; RMSE – Root Mean Square Error; MAE – Mean Absolute Error; MB – Mean Bias; O – Observation; M - Model; O_M – Observed Mean; M_M – Model Mean; SD – Standard Deviation; Units for RMSE/MAE/MB/O_M/M_M/O_SD/M_SD:  ºC for temperature, percentage for relative humidity, and m s$^{-1}$ for wind.


| Variable | Corr | RMSE | MAE | MB | O_M | M_M | O_SD | M_SD |
|---|---|---|---|---|---|---|---|---|
| Temperature | 0.74 | 7.0 | 2.8 | 0.6 | 22.2 | 22.8 | 9.5 | 8.7 |
| Relative Humidity | 0.76 | 12.1 | 9.5 | -1.1 | 67.8 | 66.6 | 14.3 | 18.6 |
| U Wind | 0.58 | 1.3 | 0.7 | 0.1 | 0.1 | 0.2 | 1.2 | 1.4 |
| V Wind | 0.49 | 1.6 | 0.7 | 0.3 | 0.2 | 0.5 | 1.6 | 1.2 |





**Table 5.** $NO_x$ emissions before and after carrying out the inversion using OMI/OMPS for different countries in May-June 2016.

| Countries | The a priori (Gg/day) | The a posteriori (Gg/day) | Changes in magnitudes | Changes in errors |
|---|---|---|---|---|
| China | 87.94±44.09[1] | 68.00±15.94[2] | -23% | -63% |
| North China Plain | 27.96±13.49 | 19.05±2.50 | -32% | -81% |
| Pearl River Delta | 4.23±1.78 | 2.70±0.32 | -36% | -84% |
| Yangtze River Delta | 9.84±4.68 | 5.77±0.51 | -41% | -89% |
| Thailand | 4.38±3.24 | 4.20±2.28 | -4% | -29% |
| Japan | 3.53±1.71 | 3.96±1.04 | +12% | -39% |
| Malaysia | 2.89±2.77 | 2.25±1.34 | -22% | -49% |
| Vietnam | 2.87±2.04 | 2.79±1.57 | -3% | -23% |
| South Korea | 2.71±1.34 | 2.95±0.58 | +9% | -56% |
| Bangladesh | 1.72±1.06 | 2.10±0.87 | +22% | -18% |
| Philippines | 1.30±1.10 | 1.54±0.98 | +18% | -11% |
| Taiwan | 1.26±0.57 | 0.97±0.33 | -23% | -42% |
| Cambodia | 0.54±0.50 | 0.57±0.45 | +5% | -11% |
| Mongolia | 0.19±0.13 | 0.28±0.12 | +44% | -8% |

1- The errors in the a priori are estimated from equation 6.
2- The errors in the a posteriori are calculated by equation 4.





**Table 6.** MDA8 surface ozone levels before and after carrying out the inversion for different regions in May-June 2016.

| Regions | The a priori (ppbv) | The a posteriori (ppbv) | Changes in magnitudes |
|---|---|---|---|
| China | 56.10±16.34 | 56.72±16.71 | +1.1% |
| North China Plain | 81.15±9.57 | 85.71±10.39 | +5.6% |
| Pearl River Delta | 65.94±9.39 | 62.37±8.93 | -5.4% |
| Yangtze River Delta | 76.79±5.90 | 82.04±5.21 | +6.8% |
| Thailand | 50.86±8.84 | 48.85±7.94 | -3.9% |
| Japan | 64.29±7.98 | 65.52±7.78 | +1.9% |
| Malaysia | 46.87±21.87 | 44.22±12.90 | -5.6% |
| Vietnam | 49.90±9.20 | 48.88±8.65 | -2.0% |
| South Korea | 84.23±3.57 | 84.90±3.69 | +0.8% |
| Bangladesh | 65.79±12.08 | 65.21±12.20 | -0.9% |
| Philippines | 27.92±9.11 | 28.69±7.92 | +2.8% |
| Taiwan | 61.55±10.88 | 54.38±8.00 | -11.6% |
| Cambodia | 39.87±3.62 | 40.20±3.46 | +0.8% |
| Mongolia | 40.11±2.52 | 40.16±2.40 | +0.1% |




Figures:

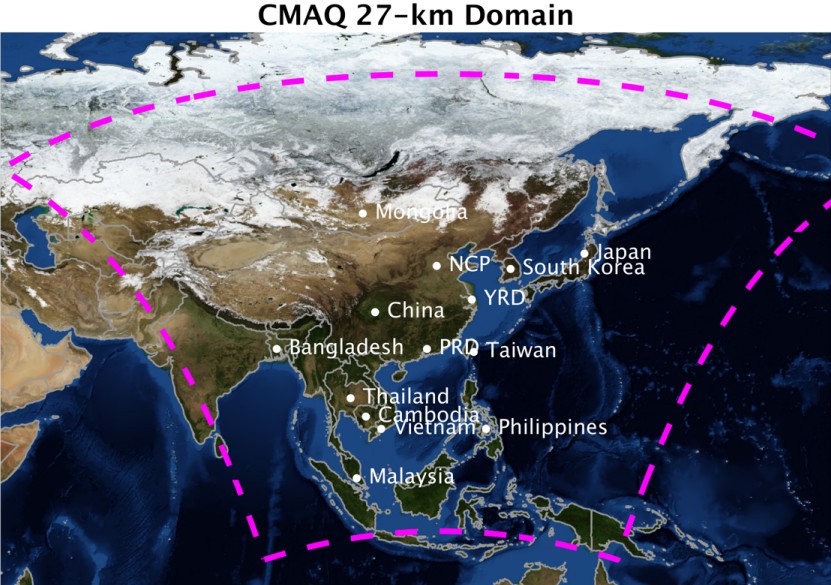

**Figure 1.** The CMAQ 27-km domain covering the major proportion of Asia. The background
890          picture is retrieved from publicly available NASA's blue marble (© NASA).



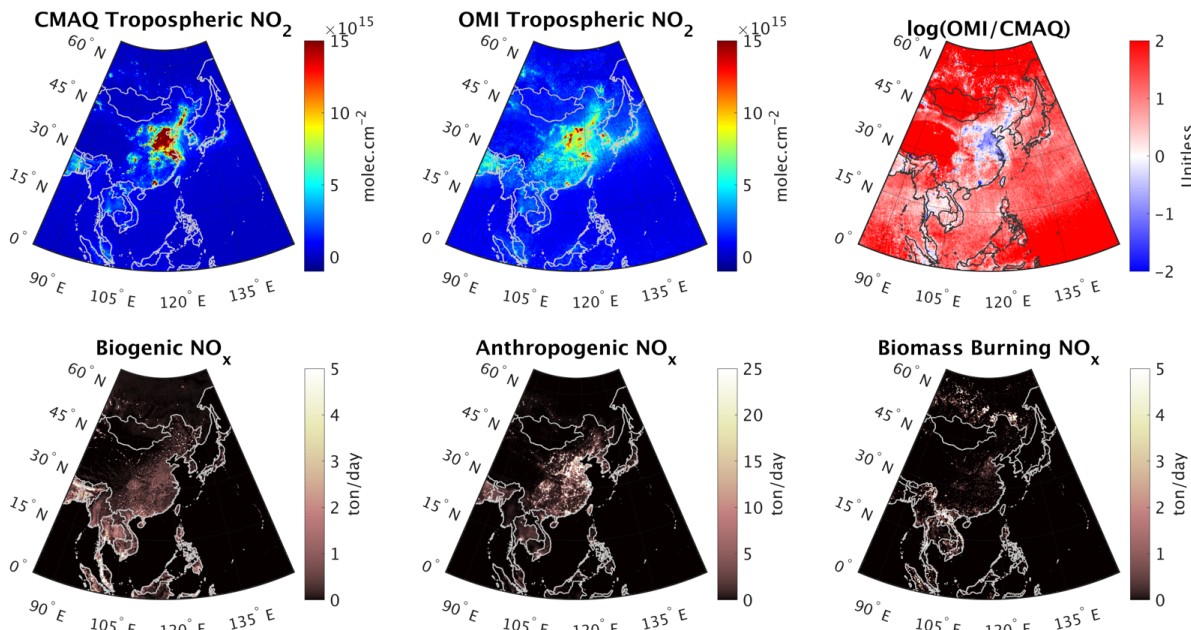

**Figure 2.** (first row), tropospheric NO$_2$ columns from the WRF-CMAQ model, OMI (using
adjusted AMFs based on the shape factors derived from the model and bias corrected following
Choi et al. [2019]), and the logarithmic ratio of CMAQ/OMI during May-June 2016 at ~1330 LST.
(second row) The major sources of NO$_x$ emissions in the region including biogenic (soil) emissions
simulated by MEGAN, anthropogenic emissions estimated by MIX Asia (2010), and biomass
burning emissions made by FINN. The emissions are the daily-mean values based on the emissions
in May-June.





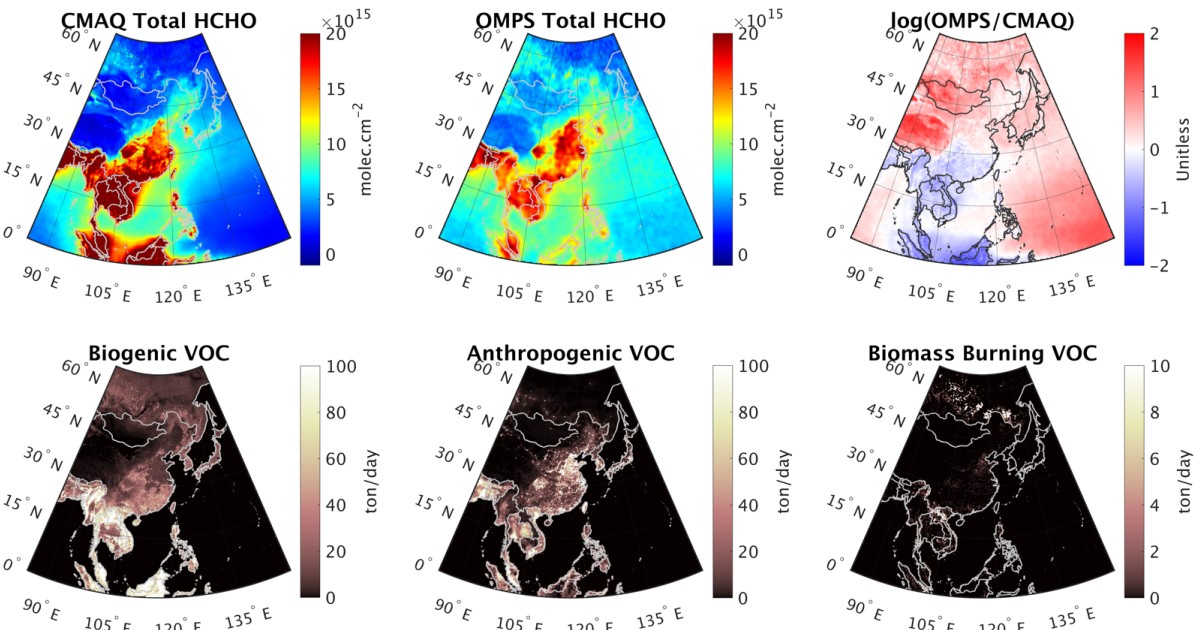

**Figure 3.** (first row), HCHO total columns from the WRF-CMAQ model, OMPS (using adjusted AMFs based on the shape factors derived from the model and bias corrected following the method proposed in Zhu et al. [2020]), and the logarithmic ratio of CMAQ/OMPS during May-June 2016 at ~1330 LST. (second row) The major sources of VOC emissions in the area including biogenic emissions simulated by MEGAN, anthropogenic emissions estimated by MIX Asia (2010), and biomass burning emissions made by FINN. The emissions are the daily-mean values based on the emissions in May-June. The VOC emissions only add up those compounds that are included in the CB05 mechanism.





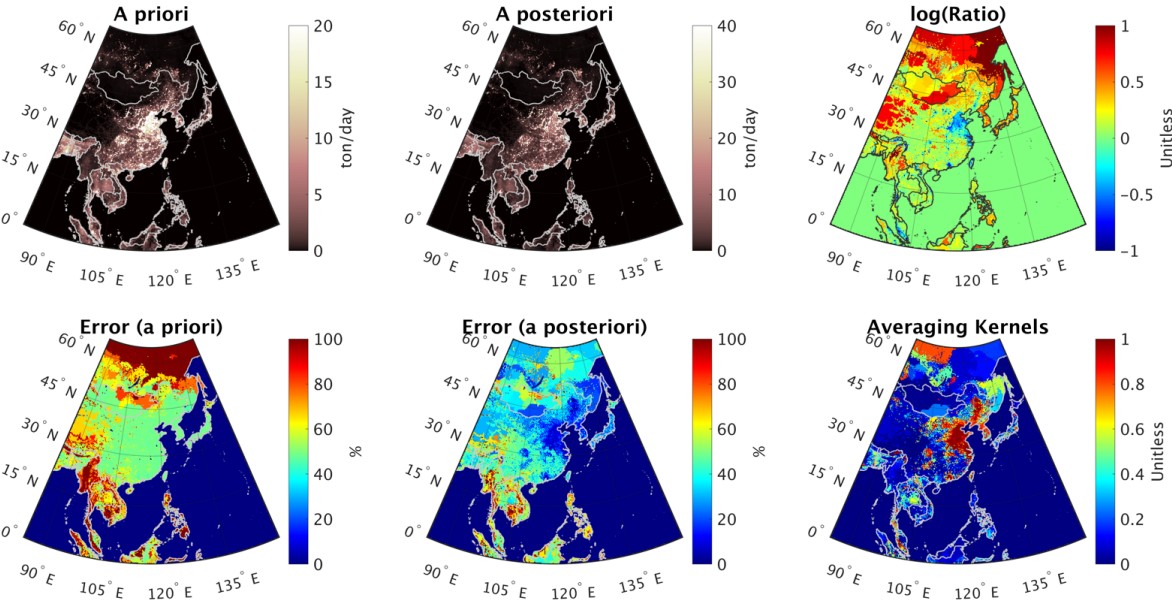

**Figure 4.** (first row), total NO$_x$ emissions (i.e., the a priori), constrained by the satellite observations (i.e., the a posteriori) in May-June 2016, and the ratio of the a posteriori to the a priori. (second row) the errors in the a priori based on Table 3, the errors in the top-down estimation, and the averaging kernels (AKs) obtained from the estimation.

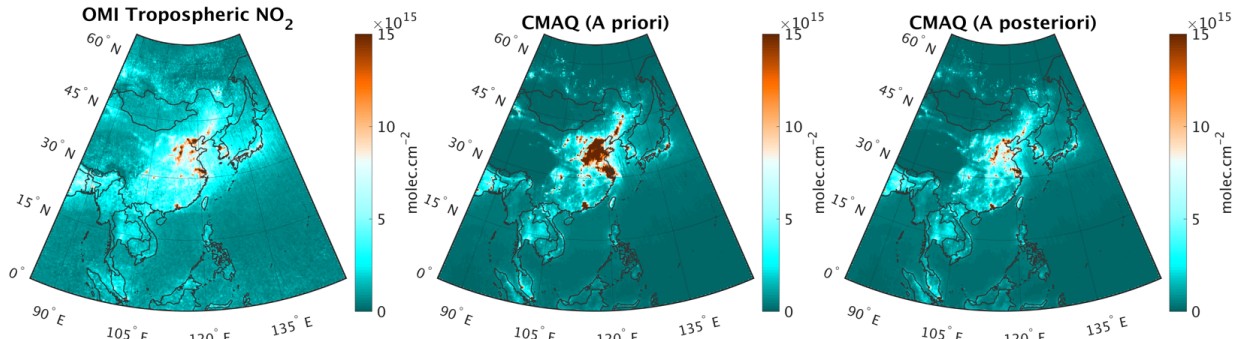

**Figure 5.** (from left to right), tropospheric NO₂ columns from OMI, WRF-CMAQ simulated with

the   prior emissions, and the same model but with the top-down emissions constrained by

OMI/OMPS in May-June 2016.



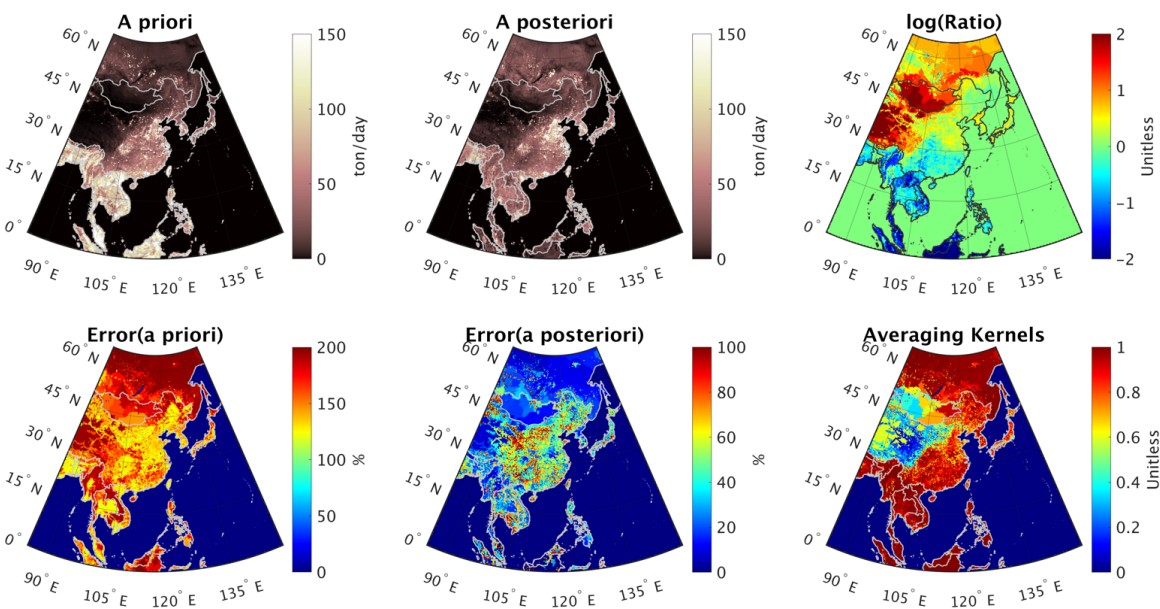

**Figure 6.** (first row), total VOC emissions (i.e., the a priori), constrained by the satellite observations (i.e., the a posteriori) in May-June 2016, and the ratio of the a posteriori to the a priori. (second row) the errors in the a priori based on Table 3, the errors in the top-down estimation, and the averaging kernels (AKs) obtained from the estimation.





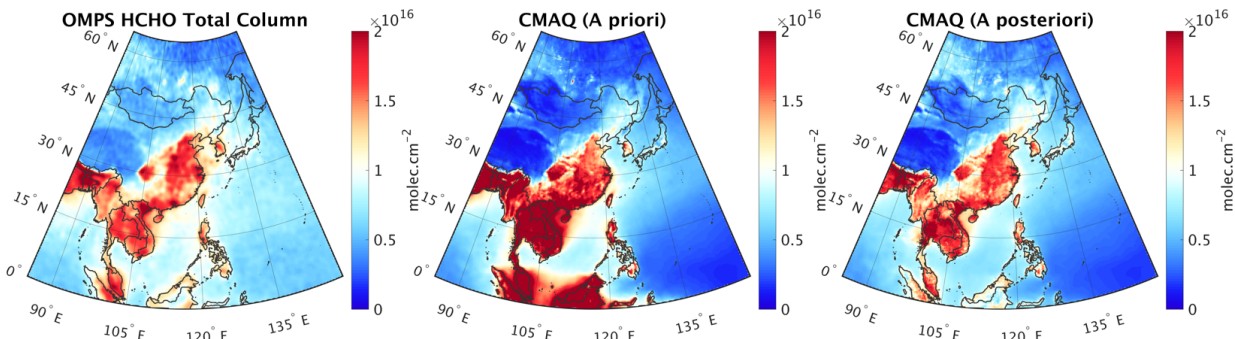

**Figure 7.** (from left to right), HCHO total columns from OMPS, the WRF-CMAQ simulated with
the prior emissions, and the same model but with the top-down emissions constrained by the
satellite in May-June 2016.

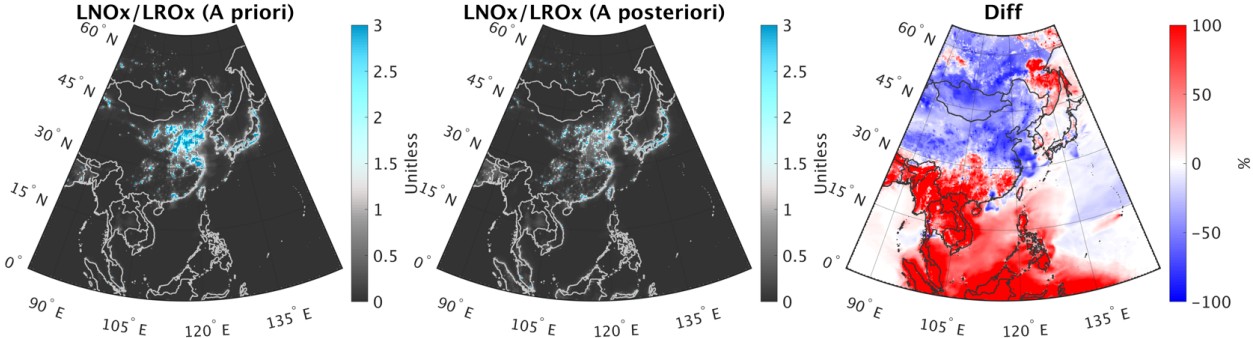

**Figure 8.** (from left to right), ratio of $LNO_x/LRO_x$ simulated by the prior and the posterior emissions,
and their differences at 1200-1800 CST, averaged over May-June 2016.





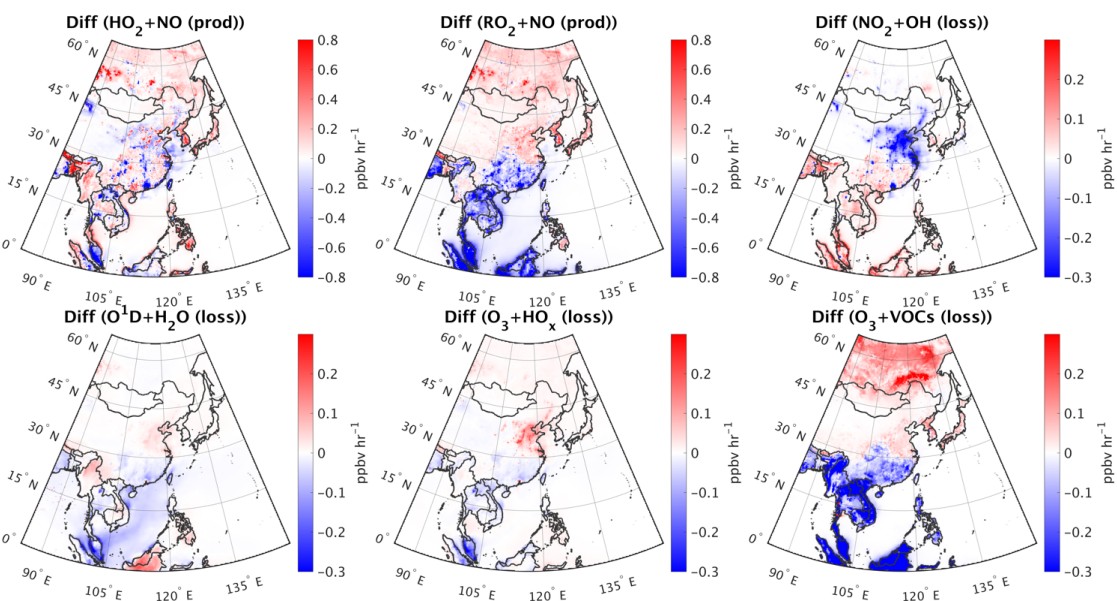

**Figure 9.** Differences between the simulations with the updated emissions and the default ones of six major pathways of ozone production/loss. The time period is May-June 2016, 1200-1800 CST.

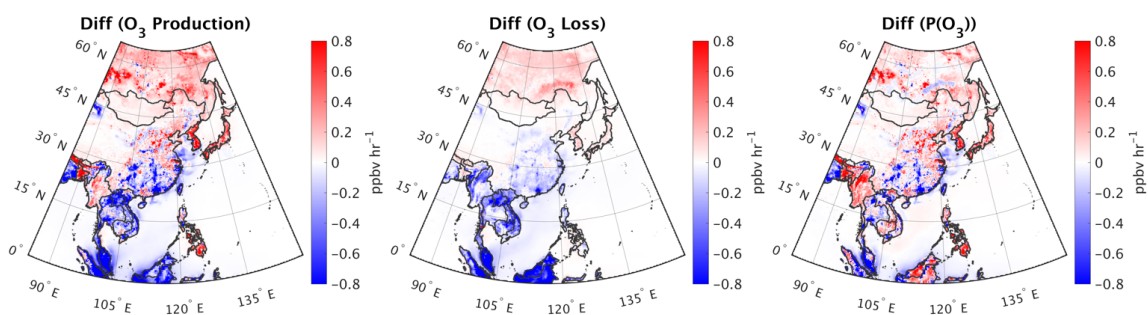

**Figure 10.** Changes in the major chemical pathways of ozone production/loss, and the net of ozone production $P(O_3)$ after updating the emissions. The time period is May-June 2016, 1200-1800 CST.





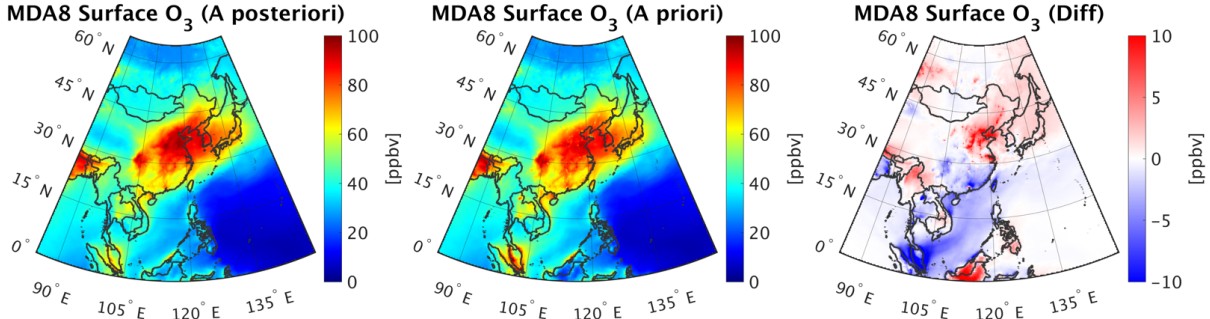

**Figure 11.** Simulated MDA8 surface ozone using the updated emissions constrained by OMI/OMPS
observations (left), the default ones (middle), and their difference (right) in May-June 2016.