# Peer review of "An Inversion of NOx and NMVOC Emissions using Satellite Observations during the KORUS-AQ Campaign and Implications for Surface Ozone over East Asia"

_Atmospheric Chemistry and Physics, 2020_

## Referee Comment (RC1) · Anonymous Referee #1 · 29 Apr 2020

Review of "An Inversion of NOx and NMVOC Emissions using Satellite Observations during the KORUS-AQ Campaign and Implications for Surface Ozone over East Asia" by Souri et al.

**General comments:**

This paper presents an optimization of $NO_x$ and VOC emissions over East Asia based on OMPS HCHO and OMI $NO_2$ during the 2016 KORUS-AQ campaign, interprets the emission changes (relative to the 2010 inventory) based on recent emission controls, and evaluates the potential impact of those emission changes on surface ozone. The use of satellite-based trace gas retrievals in a multi-species inversion is state-of-the-science, and the results are of sufficient importance to warrant publication in ACP. However, the presentation as is requires some additional work. Below is a list of specific comments that mostly denote areas where I think clarification or additional information is needed. I also think the authors could devote more space in the manuscript to the evaluation of the satellite observations and optimized simulation, including a comparison of the MDA8 $O_3$ to observations if possible. Additionally, the manuscript is quite long. I would suggest the authors consider combining/condensing the "Comparison of the model and the satellite observations" section and the "Updated Emissions" section along with the associated Figures, as there is significant overlap in the discussions between the two. The grammar could also be improved—I've noted some specific instances below where the authors should consider rewording the text, and there are plenty of other places where the language could be more concise and direct. I would recommend publication once these comments are addressed.

**Specific comments:**

Line 31: Averaging kernels themselves do not indicate whether emissions are "greatly improved" in an inversion—I would instead mention the comparisons to in situ data here.

Line 32-33: "The amount of total $NO_x$ emissions is mainly dictated by values reported in the MIX-Asia 2010 inventory." I'm not sure what this means—this inventory is used as the prior, but the results point to large decreases over much of East Asia, so surely the total $NO_x$ emissions also went down?

Lines 41-42: "We revisit the well-documented positive bias in the model in terms of biogenic VOC emissions." Can the authors be more specific here about what their results say about this positive bias?

Lines 114-116: From this, it sounds like the authors used the GEOS-Chem prediction for each specific day for the reference sector correction, rather than the climatological monthly-mean GEOS-Chem values used in Gonzales Abad (2016)? What impact does this have on the performance of the retrieval? Did the authors do any comparisons?

Line 169: The authors denote that the observational error covariance matrix corresponds to the instrument uncertainty, but model (i.e. transport) uncertainty also contributes here. Do the authors account for any model uncertainty in this term?

Line 173: "it does not allow the a posteriori to deviate largely from the a priori…" I would delete or reword this phrase, because of course this depends on how uncertain one assumes the prior emissions to be (as the authors clarify later in the paragraph).

Line 176: One question I have at the end of this paragraph is how the authors weight the relative contributions of the HCHO and $NO_2$ observations to the cost function? This seems to be an important consideration in multi-species inversions that deserves some discussion.

Line 182: Is there any metric used to support the decision to iterate three times?

Lines 192-198: Evaluation of the satellite observations with the KORUS-AQ aircraft data is a strength of this study that could use more attention in the manuscript, especially since the authors describe the satellite observations as "well-characterized". I think a figure showing the satellite-aircraft comparison would be helpful and would also serve to justify the decision to uniformly scale the HCHO and $NO_2$ columns up by the specified amounts.

Lines 227-228: "We do not consider the interconnection between the zonal emissions and concentrations due to computational burdens." I'm not clear what exactly this means. That the K matrix is assumed to be diagonal?

Lines 304-307: I found this sentence confusing. How does one determine the yield of HCHO from the OMPS data, and why does it suggest that the anthropogenic emissions dominate in NCP?

Lines 360-366: Here I think the authors are attempting to highlight the advantages of their iterative, multispecies inversion approach over simpler scaling methods, but the language is unclear and could be interpreted in the wrong way. Consider using stronger language here to show how this work advances on previous satellite-based $NO_x$ emission optimizations.

Lines 367-376: Because the OMI data are used in the inversion, this comparison is not an independent validation. I would consider moving Fig. 5 to the supplement (or perhaps combining it with Fig. 4) and focus more on the in situ comparison here.

Lines 374-376: The authors derive quite large relative changes in $NO_x$ emissions over remote regions, so it seems incorrect to say the inversion is more weighted toward the prior emissions here. Also, higher a priori error would allow for larger deviation away the prior, not toward it as the authors say. Instead, could background conditions and/or lightning sources be a significant contributor here? What does the literature say?

Lines 377-384: Why not include a figure showing the $NO_x$ comparison to aircraft data? I suggest including this comparison with or in place of the current Fig. 5.

Lines 406-412: As for $NO_x$, the HCHO validation should focus more on the in situ comparison here than on the comparison to OMPS. Consider moving Fig. 7 to the supplement (or combining it with Fig. 6).

Lines 419-428: Consider combining Figs. S2-S7 into one Figure and including it in the main text, to be referenced here.

Lines 491-504: Is there any reliable O3 data in the region to which you can compare the modeled MDA8 O3? Does the a posteriori simulation compare better to O3 measurements made during KORUS-AQ?

Lines 551-564: While it is good to highlight the remaining uncertainties and research needs at the end, this last paragraph kind of gets into the weeds in a way that ends the paper on a low note. Consider shortening this section to focus on the strengths of this study with an eye toward future improvements.

Figures 2 and 3: The caption says the upper right panels in these figures is the logarithmic ratio of model/obs, but what's actually plotted is the inverse of the ratio (obs/model). Consider replotting with the model/obs ratio, as this would be more consistent with how it is discussed in the text.

Figure 8: Can the color scale be adjusted to better indicate the values that fall above/below the transition line of 2.7?

Figures 9 and 10: Consider combining these into one Figure.

Figures S2-S7: The captions need to include some more information about what exactly is being plotted here. Are these mean profiles for the entire KORUS-AQ campaign? Was any type of filtering applied to the data?

**Technical corrections**

Throughout the manuscript: the phrase: "in terms of" is used excessively—suggest deleting it to make the discussion more concise.

Line 46: Suggest changing "an ~ 53%" to "a ~53%"

Line 51: Delete "the" before "southern China"

Lines 54-56: These sentences are a bit awkward—consider rewording.

Lines 64-71: This is a long, cumbersome sentence—consider breaking it up for better flow.

Line 69: Delete "the" before "effect"

Line 137: Delete "an" before "analytical"

Line 151: Reference should Guenther et al. (2012) instead of (2006)

Line 152: "diurnally lateral chemical conditions" should maybe be "diurnally-varying lateral chemical conditions" (?)

Lines 225-227: This description is awkward—consider rewording.

Lines 287-289: This sentence is awkward—consider rewording.

Lines 317: Change "satellites" to "satellite"

Line 320: Delete the first instance of "associated" in this sentence

Line 400: Change the word "owning" to "owing"

Line 423: Add "The" before "same tendency"

Line 447: Delete the word "condition" before "regimes"

Line 463: Insert the word "on" before "par"

Lines 472-475: The sentence is awkward—consider rewording.

Line 479: Change "forming" to "form"

Lines 486-488: The sentence is awkward—consider rewording.

---

## Referee Comment (RC2) · Anonymous Referee #2 · 22 May 2020

This manuscript performs an inversion using satellite data to estimate improvements to emission inventories of VOC's and NOx in East Asia. The research seems thorough, the results are interesting and the implications are relevant and important. I am happy to recommend publication subject to minor revisions.

Averaging Kernels are an important part of the work. They are mentioned in passing in the abstract, given a theoretical definition in the method section and then more discussion in the results. I would recommend adding a sentence in the abstract to help the non-specialist, and a more extensive explanation in the methods section to explain not

just the mathematical definition but also the physical interpretation.

In a similar vein, I felt that So and Se could be described in greater detail, especially giving more specific descriptions of the values used.

Line 258: "WRF-CMAQ largely underestimated (56%) tropospheric NO2 columns" – It would be interesting to also quote the bias in molec/cm2. CMAQ is too high in urban areas and too low in rural ones. Citing over/under predictions in molec/cm2 would give a useful perspective on some of these changes.

Minor language edits are needed throughout. For example, sometimes the text should say *the* US, *the* PRD. "representivity", "intertwisted" need correcting.

---

## Author Comment (AC1) · 1 Jul 2020

*Review of "An Inversion of NOx and NMVOC Emissions using Satellite Observations during the KORUS-AQ Campaign and Implications for Surface Ozone over East Asia" by Souri et al.*

*General comments:*

*This paper presents an optimization of $NO_x$ and VOC emissions over East Asia based on OMPS HCHO and OMI $NO_2$ during the 2016 KORUS-AQ campaign, interprets the emission changes (relative to the 2010 inventory) based on recent emission controls, and evaluates the potential impact of those emission changes on surface ozone. The use of satellite-based trace gas retrievals in a multi-species inversion is state-of-the-science, and the results are of sufficient importance to warrant publication in ACP. However, the presentation as is requires some additional work. Below is a list of specific comments that mostly denote areas where I think clarification or additional information is needed. I also think the authors could devote more space in the manuscript to the evaluation of the satellite observations and optimized simulation, including a comparison of the MDA8 $O_3$ to observations if possible. Additionally, the manuscript is quite long. I would suggest the authors consider combining/condensing the "Comparison of the model and the satellite observations" section and the "Updated Emissions" section along with the associated Figures, as there is significant overlap in the discussions between the two. The grammar could also be improved—I've noted some specific instances below where the authors should consider rewording the text, and there are plenty of other places where the language could be more concise and direct. I would recommend publication once these comments are addressed.*

**The authors are grateful for the time and thoughts this reviewer has put into his/her review.**

**We combined two sections for sake of brevity.**

**Our response follows:**

*Specific comments:*
*Line 31: Averaging kernels themselves do not indicate whether emissions are "greatly improved" in an inversion—I would instead mention the comparisons to in situ data here.*

**Thanks for the comment. Improvements can be categorized in i) a narrower uncertainty and ii) a reduction of bias, both of which are pivotal. As the reviewer #2 has pointed out, one of the strengths of this study is that it informs the amount of information gained from the observations by explicitly quantifying the averaging kernels, so we decide to keep this sentence, but modify it to:**

*"Emission uncertainties are greatly narrowed (averaging kernels>0.8, which is the mathematical presentation of the partition of information gained from the satellite observations with respect to the prior knowledge) over medium- to high-emitting areas such as cities and dense vegetation."*

**We will discuss about the comparison with in-situ data later on.**

*Line 32-33: "The amount of total $NO_x$ emissions is mainly dictated by values reported in the*

*MIX-Asia 2010 inventory.” I’m not sure what this means—this inventory is used as the prior, but the results point to large decreases over much of East Asia, so surely the total NO$_x$ emissions also went down?*

**Thanks for your detailed comment. We added:** *“the prior amount of ...”*

*Lines 41-42: “We revisit the well-documented positive bias in the model in terms of biogenic VOC emissions.” Can the authors be more specific here about what their results say about this positive bias?*

**We found that MEGAN v2.1 estimated too much isoprene in tropics. We added the factor of overestimation and the name of model:**

*“We revisit the well-documented positive bias (by a factor of 2 to 3) of the MEGAN v2.1 in terms of biogenic VOC emissions in the tropics.”*

*Lines 114-116: From this, it sounds like the authors used the GEOS-Chem prediction for each specific day for the reference sector correction, rather than the climatological monthly-mean GEOS-Chem values used in Gonzales Abad (2016)? What impact does this have on the performance of the retrieval? Did the authors do any comparisons?*

**To investigate to what degree the prior profiles affect the retrieval, we compared and added the following figure in the supplement and wrote. Please note these results are not corrected for shape factors and biases (shouldn’t change the difference).**

[Figure]

*Figure S1. A comparison of the impact of the reference correction on the amount of HCHO total columns (not corrected for shape factors and systematic biases). Daily and monthly denote that the OMPS HCHO vertical columns were computed using the daily and the monthly means (2004-2017) of the GEOS-Chem profiles, respectively. The difference is about 4% on average.*

*“An upgrade to this reference correction is the use of daily HCHO profiles over monthly-mean climatological ones from simulations done by the GEOS-Chem chemical transport model. On average, this leads to a 4% difference in HCHO total columns with respect to using only the monthly-mean climatological values (Figure S1).”*

*Line 169: The authors denote that the observational error covariance matrix corresponds to the instrument uncertainty, but model (i.e. transport) uncertainty also contributes here. Do the*

*authors account for any model uncertainty in this term?*

**Unfortunately not, propagating the model error parameters (such as winds, PBL, clouds and etc.) to the final estimation requires a fully explicit calculation of Jacobians (here linking columns to that specific parameter) which is computationally burdensome. That's an oversight which we had touched upon in the conclusion part later. Concerning random errors in the model (which is mostly caused by numerical diffusion and discontinuities), one may estimate them using the NMC method (commonly used in the weather data assimilation area, e.g., https://doi.org/10.1016/j.atmosres.2020.104965) that again requires a lot of perturbations/predictions. Those values may have been significantly reduced by oversampling, but the model error parameters (which won't be averaged) are indeed important. So, we essentially tend to under-predict the errors in the top-down estimation because of treating the model parameters as perfect.**

*Line 173: "it does not allow the a posteriori to deviate largely from the a priori…" I would delete*
*or reword this phrase, because of course this depends on how uncertain one assumes the prior emissions to be (as the authors clarify later in the paragraph).*

**Thanks, we removed it.**

*Line 176: One question I have at the end of this paragraph is how the authors weight the relative contributions of the HCHO and NO2 observations to the cost function? This seems to be an important consideration in multi-species inversions that deserves some discussion.*

**That's a very neat question. Many levels of sophistication exist when it comes to implementing a joint inversion framework. A very simple way would be by separately incorporating HCHO and NO2. The major problem with this way is the complexities associated with the chemical non-linearities; NOx and VOC impact their own concentrations. Here, the order becomes important, meaning it would be different if we constrained NOx first and then VOC and vice versa. The second way is to explicitly consider the cross-relationships (i.e., the derivatives of HCHO to NOx and NO2 to VOC shown in color fonts) (from our AGU's poster):**

[Figure]

The other way is to ignore those cross-relationships in Jacobian, and perform a non-linear analytical inversion (Gauss-Newton) iteratively, so the chemical feedback will be implicitly (and incrementally) passed on to the main derivatives (NO$_2$-NO$_x$ and HCHO-VOC). We tested both approaches, and we came up with a conclusion that the latter is more robust, especially for our case when two different sensors were used (it wasn't smart to have co-registered cross-relationships between OMI and OMPS, for example, in row anomaly situations happening in OMI, we had to also remove the same pixels from OMPS to look at the same footprint meaning that we would have sacrificed OMPS information for OMI).

The other important part is how we go about the covariance matrix of observations which partly addresses this reviewer's question. We did not consider non-diagonal values meaning the weight of each specie is dictated by its own error. For instance, the HCHO would have higher weight compared to NO2 over rural/vegetated areas. One may argue that this is a not complete joint inversion because we did not consider co-variances. We speculate that the translation of the covariance matrix of observations to the emission space is mainly achieved by the Kalman gain (G) which has been estimated iteratively by information coming from both species. So, the way errors are propagating in the inversion keeps up with the non-linearities that are considered in our work.

We believe interconnectedness is a core characteristic of atmospheric composition and yet is frequently ignored in the area of inverse modeling and data assimilation. To consider the tangled relationships between atmospheric compounds such as the potential effect of oxidation and lifetime of one on another, we should utilize a proper optimizer and estimate gradients incrementally (both of which were tackled in this study).

To account for the reviewer's comment:

*"This error is based on the RMSE obtained from the mentioned studies used for removing biases. Despite the fact that we do not account for non-diagonal elements of the covariance matrices, the incremental updates of **G** adjusted by both NO$_2$ and HCHO observations should better translate the covariance matrices into the emission space."*

*Line 182: Is there any metric used to support the decision to iterate three times?*

The number of iterations were set purely based on the computation/time limitations. We did not use any threshold as criterion. It is worth noting that due to the nature of the analytical inversion, these calculations were all done offline which were very labor-intensive.

*Lines 192-198: Evaluation of the satellite observations with the KORUS-AQ aircraft data is a strength of this study that could use more attention in the manuscript, especially since the authors describe the satellite observations as "well-characterized". I think a figure showing the satellite-aircraft comparison would be helpful and would also serve to justify the decision to uniformly scale the HCHO and NO2 columns up by the specified amounts.*

**Thanks for your suggestion. We now included the comparison of GEOS-Chem (corrected with KORUS-AQ data) and OMPS HCHO (adjusted for shape factors using the WRF-CMAQ model) in the supplementary [Figure S2]:**

[Figure]

*Figure S2. The comparison of the corrected GEOS-Chem model using DC8 observations during the KORUS-AQ campaign (left), and OMPS HCHO columns (corrected for shape factors) (right). The method is fully described in Zhu et al. [2016; 2020].*

**Regarding NO₂, the comparisons have already been discussed in Choi et al. [2019]. We simply used their results.**

*Lines 227-228: "We do not consider the interconnection between the zonal emissions and concentrations due to computational burdens." I'm not clear what exactly this means. That the K matrix is assumed to be diagonal?*

**To reduce the size of K matrix, we grouped the region into certain zones using the GMM method. More than 10k zones were labeled, some as small as a grid cell, others can get as large as a country (such as Mongolia). The inversion was done separately for each zone (two emissions, sum of NOx and sum of VOC), and as many OMPS/OMI observations as available within the zone. We did not consider the impact of one zone to another one (no source-receptor relationship outside of a zone). Yes, if we look at the full K matrix (all zones together), there will be many zeros (can't say fully diagonal from a mathematical standpoint). To fill up those values, we have to run the forward model more than 10k times 2 (NOx and VOC) which is obviously not feasible.**

**We believe the HCHO and NO₂ concentrations are mostly confined to their sources in the two-month averages. One reasonable way to implicitly consider the source-receptor relationship (aka, transport) is to numerically solve the optimization using the adjoint of the model (which unfortunately has not been updated for years). We added:**

*"We do not consider the interconnection between the zonal emissions and concentrations due to computational burdens; therefore, we assume that the HCHO and NO$_2$ columns are mostly confined to their sources in the two-month averages."*

*Lines 304-307: I found this sentence confusing. How does one determine the yield of HCHO from the OMPS data, and why does it suggest that the anthropogenic emissions dominate in NCP?*

**Sorry for using the wrong term. We changed it to "***the concentrations of***".**

*Lines 360-366: Here I think the authors are attempting to highlight the advantages of their iterative, multispecies inversion approach over simpler scaling methods, but the language is unclear and could be interpreted in the wrong way. Consider using stronger language here to show how this work advances on previous satellite-based NO$_x$ emission optimizations.*

**Thanks. We modified the paragraph and added:"** *ii) the CMAQ-DDM (Figure S3) suggests that NO$_2$ columns decrease due to increasing VOC emissions over the region; accordingly, the cross-relationship between NO$_2$ concentrations and VOC emissions partly adds to the discrepancy. It is because of the chemical feedback that recent studies have attempted to enhance the capability of inverse modeling by iteratively adjusting relevant emissions [e.g., Cooper et al., 2017; Li et al., 2019]. Likewise, our iterative non-linear inversion shows a superior performance over traditional bulk ratio methods, in part because it considered incrementally the chemical feedback."*

*Lines 367-376: Because the OMI data are used in the inversion, this comparison is not an independent validation. I would consider moving Fig. 5 to the supplement (or perhaps combining it with Fig. 4) and focus more on the in situ comparison here.*

**We fully understand the reviewer's concern. There are a couple of reasons that we initially decided to move the independent comparisons to the supplement:**

   i)     **There are not significant changes in emissions over South Korea (they are mostly spread out). The noticeably concentrated change is over Seoul. Unfortunately, the KORUS-AQ campaign DC-8 measurements suffer from the lack of frequent spiral measurements (there was no single spiral measurement over Seoul during the whole campaign). This means the majority of observations sampled in places where we did not really see a major change (either there shouldn't be much change, or the satellite observation didn't have adequate temporal/spatial information to induce a change).**

   ii)    **The inversion was done in a course of two-month average, whereas the DC-8 observations have sporadic measurements around the Korean Peninsula. So, it's unfair to ask from the model to reproduce those observations, because we did not guide the model with high temporal information.**

iii) **As someone who performs inverse modeling off and on, we always ask ourselves if looking at concentrations is a concrete way of validating top-down emissions. Concentrations can be impacted by other variables that are not constrained in the model. It is quite possible that many underlying errors in the model result in a seemingly reasonable output (right for a wrong reason), therefore, improving separately a component would make the result seem worse. A worse result after the adjustment could be actually promising because the new adjustment is bringing out other issues in the model that had been wrongly canceled out in the beginning. It is worth noting that our inversion either improved the results compared to the independent measurements or remained in the same error range. One may say, our previous studies (Souri et al., 2016; Souri et al., 2017a; Souri et al., 2018; Souri et al., 2020a: https://doi.org/10.1029/2019JD031941, Souri et al., 2020b: https://doi.org/10.1016/j.atmosres.2020.104965) assessed changes in concentration (or other diagnostic variables) and used them as evidence of improvement, so why are we cherry-picking? Those studies focus on very drastic changes, so the off emissions (or other prognostic variables) dominated over unresolved model/observational errors. This is not the case for the KORUS-AQ campaign over South Korea in the two-month averages.**

iv) **We strongly believe the only way to validate top-down emission is by looking at the flux observations measured by eddy covariance or CAMS flux measurements (apples-to-apples).**

v) **Checking the constrained model with the used observations (internal validation, or control points) is as important as looking at independent measurements (benchmarks). It goes to show that the inversion framework is not faulty.**

**Having said that, we keep the independent measurements in the supplementary.**

*Lines 374-376: The authors derive quite large relative changes in $NO_x$ emissions over remote regions, so it seems incorrect to say the inversion is more weighted toward the prior emissions here. Also, higher a priori error would allow for larger deviation away the prior, not toward it as the authors say. Instead, could background conditions and/or lightning sources be a significant contributor here? What does the literature say?*

**Thanks for your comment. Yes, we used a wrong sentence for this part. We removed the sentence. OMI/CMAQ ratio suggests that we should increase NOx emissions by a factor of 10 over remote areas; such value is not supported by the inversion. This is because the observation covariance is large compared to the absolute value of columns in remote areas, in a relative sense. To account for the reviewer's comment, we removed the sentence and added:**

*"However, we do not see a significant change in the background values in the new simulation compared to those of OMI due to less certain column observations."*

**Regarding the reasons for the low background conditions, we already had mentioned some speculations about the problem, but we want to empathize that tackling the model issues by**

looking at satellite observations whose columns are biased-high in rural areas and possess relatively large errors (weaker signals) is overrated. Likewise, the uncertainties associated with top-down lightning NOx from satellites are large (>60%) [Allen et al., 2019; https://doi.org/10.1029/2019JD030561] mainly due to the assumptions made for cloud optical impacts on the scattering weights. However, there are some promising studies looking at nitrate family challenges such as [Romer Present et al., 2020] that may indirectly address some model issues.

*Lines 377-384: Why not include a figure showing the NOx comparison to aircraft data? I suggest including this comparison with or in place of the current Fig. 5.*

**We already discussed about this in the above comments. We now included the figure in the supplementary:**

[Figure]

***Figure S4***. Comparison of the simulated model using the prior/posterior emissions and DC-8 measurements in terms of NO₂ mixing ratios. We included all 10-secs observations available from DC-8 four-channel NCAR's chemiluminescence in May-June 2016. The profiles are the mean average.

*Lines 406-412: As for NOx, the HCHO validation should focus more on the in situ comparison here than on the comparison to OMPS. Consider moving Fig. 7 to the supplement (or combining it with Fig. 6).*

**Discussed before.**

*Lines 419-428: Consider combining Figs. S2-S7 into one Figure and including it in the main text to be referenced here.*

**Discussed before.**

*Lines 491-504: Is there any reliable O3 data in the region to which you can compare the modeled MDA8 O3? Does the a posteriori simulation compare better to O3 measurements made during KORUS-AQ?*

**Looking at Chinese surface O3 observations (where the major change in concentration occurs), we do see the simulation become better at some regions (southern parts) and worse at others (northern parts). This by no means should be used to undermine the quality of the inversion, as the CTMs tend to largely overpredict surface ozone due to multiple reasons (predominantly vertical mixing, too transparent clouds (inability of the model to capture deep convections), chlorine chemistry and large bias associated with global emissions) [Travis et al., 2016]. We now included the MDA8 surface ozone in the paper, but added some caveats about preexisting issues in the CTM models. At least, our study shows that other underlying issues are more important compared those of emissions, a finding which is line with previous studies:**

[Figure]

**Figure 11.** Simulated MDA8 surface ozone using the updated emissions constrained by OMI/OMPS observations (left), the default ones (middle), and their difference (right) in May-June 2016. We overplot surface MDA8 ozone values (circles) from the Chinese air quality monitoring network (https://quotsoft.net/air/).

*"Comparisons with surface observations show that the model generally captured the ozone spatial distributions; however, it tends to largely overpredict MDA8 surface ozone (~ 7 ppbv). This tendency has been well-documented in other studies [e.g., Travis et al., 2016; Souri et al., 2017b; Lu et al., 2019]. The updated simulation with the top-down emission partly reduces this overestimation in the southern regions of China, while it further exacerbates the overestimation in the northern parts. No doubt much of this stems from the fact that the preexisting biases associated with the model (beyond emissions such as vertical mixing and cloud optical thickness) mask any potential improvement expected from the constrained emissions. Because of this, in*

*addition to adjusting relevant emissions, a direct assimilation of ozone concentrations should complimentarily be exploited [e.g., Miyazaki et al., 2019] to bolster the capability of the model at simulating ozone."*

**In conclusion:**

*"The comparison of simulated ozone before and after adjusting emissions and Chinese surface air quality observations reveal a large systematic positive bias (~ 7 ppbv) which hinders attaining the benefits from a more accurate ozone production rate due to the observationally-constrained $NO_x$/VOC ratios. This highlights the need to explicitly deal with other underlying issues in the model [e.g., Travis et al., 2016] to be able to properly simulate surface ozone."*

*Lines 551-564: While it is good to highlight the remaining uncertainties and research needs at the end, this last paragraph kind of gets into the weeds in a way that ends the paper on a low note. Consider shortening this section to focus on the strengths of this study with an eye toward future improvements.*

**Thanks, we have shortened this part and finished the paper with a higher pitch:**

*"Despite these limitations, this research demonstrated that a joint inversion of $NO_x$ and VOC emissions using well-characterized observations significantly improved the simulation of HCHO and $NO_2$ columns, permitting an observationally-constrained quantification of the response of ozone production rates to the emission changes."*

*Figures 2 and 3: The caption says the upper right panels in these figures is the logarithmic ratio of model/obs, but what's actually plotted is the inverse of the ratio (obs/model). Consider replotting with the model/obs ratio, as this would be more consistent with how it is discussed in the text.*

**Thanks, we preferred to change the text. (all now are obs/model).**

*Figure 8: Can the color scale be adjusted to better indicate the values that fall above/below the transition line of 2.7?*
**Thanks, we tried to set the center of color scale at 2.7, but that would leave majority of areas gray. Another way to use log(x) which makes the difference a bit confusing. So we decided to leave the figure with a minor change.**

*Figures 9 and 10: Consider combining these into one Figure.*
**Thanks, it would become too large.**

*Figures S2-S7: The captions need to include some more information about what exactly is being*

*plotted here. Are these mean profiles for the entire KORUS-AQ campaign? Was any type of filtering applied to the data?*

**Thanks, we added in the caption that we included all 10-secs observations during the entire campaign. The data had already gone through some tests/revision.**

*Technical corrections*
*Throughout the manuscript: the phrase: "in terms of" is used excessively—suggest deleting it to make the discussion more concise.*
*Line 46: Suggest changing "an ~ 53%" to "a ~53%"*
**Corrected.**
*Line 51: Delete "the" before "southern China"*
**Corrected.**
*Lines 54-56: These sentences are a bit awkward—consider rewording.*
**Thanks, we changed it to:**
*"Simulations using the updated emissions indicate* increases in maximum daily 8-hour average (MDA8) surface ozone over China (0.62 ppbv), NCP (4.56 ppbv), and YRD (5.25 ppbv), *suggesting that emission control strategies on VOCs should be prioritized to curb ozone production rates in these regions."*
*Lines 64-71: This is a long, cumbersome sentence—consider breaking it up for better flow.*
**Shortened.**
*Line 69: Delete "the" before "effect"*
**Corrected.**
*Line 137: Delete "an" before "analytical"*
**Corrected.**
*Line 151: Reference should Guenther et al. (2012) instead of (2006)*
**Corrected.**
*Line 152: "diurnally lateral chemical conditions" should maybe be "diurnally-varying lateral chemical conditions" (?)*
**Corrected.**
*Lines 225-227: This description is awkward—consider rewording.*
**Corrected to** *:*
*"where $S_{(1,1)}^{NO2}$ is the DDM output in units of molecule cm$^{-2}$ for the first row and column. It explains the resultant change in NO$_2$ column by changing one unit of total NO$_x$ emissions."*

*Lines 287-289: This sentence is awkward—consider rewording.*
**Thanks, it was really bad. We corrected it to** *"Accordingly, future improvements in physical/chemical processes of models will offset top-down emission estimates, inevitably."*
*Lines 317: Change "satellites" to "satellite"*
**We removed this sentence for shortening the paper.**
*Line 320: Delete the first instance of "associated" in this sentence*
**Thanks removed.**
*Line 400: Change the word "owning" to "owing"*
**Corrected.**
*Line 423: Add "The" before "same tendency"*
**Corrected.**
*Line 447: Delete the word "condition" before "regimes"*

**Corrected.**

*Line 463: Insert the word "on" before "par"*

**Corrected.**

*Lines 472-475: The sentence is awkward—consider rewording.*

**Thanks, reworded:** *The ozone photolysis ($O^1D+H_2O$) are majorly driven by photolysis and water vapor mixing ratios, both of which are roughly constant in both simulations; accordingly the difference map of $O^1D+H_2O$ is mainly reflecting changes in ozone concentrations (shown later).*

*Line 479: Change "forming" to "form"*

**Corrected.**

*Lines 486-488: The sentence is awkward—consider rewording.*

**Thanks, corrected to** *"In general, the differences in $P(O_3)$ follow the changes in the $NO_x$ emissions depending on which chemical regimes prevail."*

**The reviewer provided very detailed and constructive comments which we have taken to heart when revising the paper. We believe our paper has become stronger as a result, and hope this reviewer will find it publishable for ACP.**

---

## Author Comment (AC2) · 1 Jul 2020

*This manuscript performs an inversion using satellite data to estimate improvements to emission inventories of VOC's and NOx in East Asia. The research seems thorough, the results are interesting and the implications are relevant and important. I am happy to recommend publication subject to minor revisions.*

**Thanks for your review and recommending a minor revision.**

*Averaging Kernels are an important part of the work. They are mentioned in passing in the abstract, given a theoretical definition in the method section and then more discussion in the results. I would recommend adding a sentence in the abstract to help the non-specialist, and a more extensive explanation in the methods section to explain not just the mathematical definition but also the physical interpretation.*

**Thanks for your comment, we added:** *"Emission uncertainties are greatly narrowed (averaging kernels>0.8, which is the mathematical presentation of the partition of information gained from the satellite observations with respect to the prior knowledge) over medium- to high-emitting areas such as cities and dense vegetation."*

*In a similar vein, I felt that So and Se could be described in greater detail, especially giving more specific descriptions of the values used.*

**Thanks, we added the following details quantifying different components of the covariance matrix:**

*"We calculate the covariance matrix of observations using the column uncertainty variable provided in the satellite datasets and consider them as random errors associated with spectrum fitting. We consider 25% random errors for air mass factor calculations. Therefore, these values (as random errors) are significantly lowered down by oversampling the data over the course of two months. In addition to that, we consider a fixed error for all pixels due to variability that exists in the applied bias correction ($3.61 \times 10^{15}$ molec.cm$^{-2}$ for $NO_2$ and $4.62 \times 10^{15}$ molec.cm$^{-2}$ for HCHO). This error is based on the RMSE obtained from the mentioned studies used for removing biases. Despite the fact that we do not account for non-diagonal elements of the covariance matrices, the incremental updates of $G$ adjusted by both $NO_2$ and HCHO observations should better translate the covariance matrices into the emission space."*

*Line 258: "WRF-CMAQ largely underestimated (56%) tropospheric NO2 columns" – It would be interesting to also quote the bias in molec/cm2. CMAQ is too high in urban areas and too low in rural ones. Citing over/under predictions in molec/cm2 would give a useful perspective on some of these changes.*

**Thanks, we now added the molec/cm2 values too.**

*Minor language edits are needed throughout. For example, sometimes the text should say \*the\* US, \*the\* PRD. "representivity", "intertwisted" need correcting.*

**Corrected.**